



# Mesoscale atmospheric circulation controls of local meteorological elevation gradients on Kersten Glacier near Kilimanjaro summit

Thomas Mölg[1], Douglas R. Hardy[2], Emily Collier[1], Elena Kropač[1], Christina Schmid[1], Nicolas J. Cullen[3], Georg Kaser[4], Rainer Prinz[4], Michael Winkler[5]

[1]Climate System Research Group, Institute of Geography, Friedrich-Alexander-University Erlangen-Nürnberg (FAU), Germany
[2] Department of Geosciences, University of Massachusetts, Amherst, USA
[3] School of Geography, University of Otago, Dunedin, New Zealand
[4] Department of Atmospheric and Cryospheric Sciences (ACINN), University of Innsbruck, Austria
[5] Zentralanstalt für Meteorologie und Geodynamik (ZAMG), Austria

*Correspondence to*: Thomas Mölg (thomas.moelg@fau.de)

**Abstract.** Elevation gradients of meteorological variables in mountains are of interest to a number of scientific disciplines and often required as parameters in modeling frameworks. Measurements of such gradients on glaciers, however, are particularly scarce and strongly skewed towards the mid latitudes and valley glaciers. This article adds a tropical perspective and presents four years of overlapping measurements at 5603 m and 5873 m on Kersten Glacier, Kilimanjaro (East Africa), between 2009 and 2013. Mean gradients in near-surface air temperature ($T$), water vapor pressure (VP) and snow accumulation (ACC) per 100 m elevation are –0.75 °C, –0.16 hPa and –114 ± 16 mm w.e. year$^{-1}$, respectively. An intriguing feature is a strong diurnal cycle of the $T$ and VP gradients, which are (depending on season) 2–4 times larger between early and late morning than in the hours of weak gradients. The ACC decrease with elevation, furthermore, is mostly the result of a lower frequency of ACC events at the upper measurement site and not due to contrasting amounts at the two altitudes during events. A novel facet of our study is to link the measured on-glacier gradients to a high-resolution atmospheric modeling data set, which reveals the importance of the mesoscale atmospheric circulation. A thermally direct circulation is established over the mountain in response to diabatic surface heating/cooling with upslope flow during the day and downslope flow in the night. This persistent circulation communicates heat and moisture changes in the lowlands to the higher elevations during morning and early afternoon, which is evident in the advection patterns of potential temperature and VP, and shapes the time-variability of gradients as recorded by our weather stations on the glacier. A few local processes seem to matter as well (glacier sublimation, turbulent heat fluxes), yet they show a secondary influence only during limited time windows. Atmospheric model data also demonstrate that declining moist entropy and water vapor fluxes in the summit zone favor formation of the negative ACC gradient. The results extend the empirical basis of elevation gradients in high mountains, and in particular over glacier surfaces, by the unusual case of a slope glacier on an equatorial, free-standing massif. Our measurement/model link, moreover, demonstrates an approach for future studies to put observations of elevation gradients more systematically in a multiscale process context.



## 1 Introduction

As the world's mountain glaciers are shrinking increasingly fast, employment and development of glacier-wide mass balance models has intensified over the past decade. These models were realized for the local (e.g., Klok and Oerlemans, 2002; Mölg et al., 2009a), regional (e.g., Machguth et al., 2009; Farinotti et al., 2015), and global scales (e.g., Radic and Hock, 2011; Marzeion et al., 2012). The cited studies are only a few examples and, like the majority of the related research, they range in model structure from simple, parsimonious approaches to complex architectures – a choice driven by both the considered space/time scales and data availability. A unifying element in all these models, however, is the inclusion of elevation gradients in basic meteorological variables as model parameters. These parameters are needed to construct spatially distributed fields of the atmospheric drivers of glacier mass balance, unless the mass balance model is coupled to an atmospheric model that delivers these drivers at the different elevations (e.g., Collier et al., 2013).

While this coupling is quite common for ice sheets (e.g., Vizcaíno et al., 2010), it is still less practical for mountain glaciers due to spatial resolution issues and the associated computational expense (Mölg and Kaser, 2011). Mass balance models have therefore continued to rely on gradient parameters. Most often these concern (near-) surface air temperature and precipitation, where the mean (mid-latitude) lapse rate of $-6.5$ K km$^{-1}$ and available precipitation data have been widely employed. If air humidity is another model input, the assumption of constant relative humidity with elevation has prevailed. Information on gradients can also be gained from numerical atmospheric models (e.g., Mölg and Scherer, 2012) or high-resolution satellite products (e.g., González and Garreaud, 2017), if they resolve the mountain atmosphere adequately. Since that requirement involves challenges (e.g., model setups, retrieval algorithms, and record length), there is however no question that ground measurements are the centerpiece for advancing our understanding of the given problem.

In the context of whole mountains, the literature body on observations of elevation gradients is massive. Due to their importance in different scientific disciplines, however, the relevant information "is widely scattered" (Barry, 1992) and not restricted to atmospheric studies. For example, relatively dense station networks for observing weather at different mountain elevations were initiated in connection with glaciological (e.g., Shea et al., 2015), hydrological (e.g., Buytaert et al., 2006), and ecological questions (e.g., Hemp, 2006; Patsiou et al., 2017). Such studies primarily utilized the observed gradients, while similar field experiments in the climatology-orientated literature also served as a basis to access a sense of the possible long-term patterns and underlying processes (e.g., Pepin et al., 2010). The meteorology literature, in turn, exhibits a focus on the processes. Measurements of vertical gradients were initialized and analyzed in the context of regional atmospheric flows (e.g., Mayr et al., 2003) and associated orographic precipitation (e.g., Smith et al., 2009), air pollution problems (e.g., Gohm et al., 2009), and for understanding the differences in atmospheric properties between mountain slopes and the surrounding troposphere (e.g., Dreiseitl, 1998; Minder et al., 2010). What all these studies demonstrated reliably is the strong spatial variability of mountain weather and climate, which stems from processes of the mountain/atmosphere interaction, for





example cold-air pooling in valleys (Daly et al., 2010) or along-slope effects of forced air mass ascent (Minder et al., 2011). While the above examples are again selective, the importance of elevation gradients is shown by their natural inclusion in review-type articles (e.g., Buytaert et al., 2011) and text books (e.g., Barry, 1992).

Measurements of elevation gradients in the atmospheric surface layer over mountain glaciers are much rarer. Both logistical
challenges and the difficult physical access are obvious reasons, and any initiatives that targeted the installation of more than one weather station and/or additional instrumentation at different altitudes are limited to glaciers with relatively easy access (e.g., Petersen and Pellicciotti, 2011). Besides case studies (e.g., Carturan et al., 2015), only a few glaciers have appeared prominently in the literature in the empirical gradient context: Pasterze (Austria) as the focus of the PASTEX field experiment (e.g., Greuell et al., 1997; Greuell and Böhm, 1998), as well as Juncal Norte (Chile), Haut d'Arolla
(Switzerland), and Miage (Italy) (e.g., Petersen and Pellicciotti, 2011; Petersen et al., 2013; Ayala et al., 2015; Shaw et al., 2016). The prime interest in such studies were gradients in air temperature (e.g., table 1 in Petersen and Pellicciotti, 2011), and thus we know that these gradients are neither constant with elevation, nor obey the temperature field over non-glaciated surfaces at the same elevation. These findings raised awareness about the limited representation of temperatures measured off-glacier for glaciological studies (e.g., Greuell and Böhm 1998) and thus weaken the mean lapse rate assumption for
modeling. Most field experiments on elevation gradients, however, were conducted in the mid latitudes and on valley-type glaciers. Hence, there is a lack of information on observed elevation gradients for tropical glaciers and other glacier types.

The present study extends the empirical basis through data from Kersten Glacier, a slope glacier on Kilimanjaro close to the equator. This glacier has been the focus of an extensive research project[1] on the climatological causes of glacier loss on
Africa's highest mountain (e.g., Cullen et al., 2006, 2013; Mölg et al., 2008, 2009a, 2009b, 2012), see further in Sect. 2.1. Even if studies of mountain climates in East Africa are limited in comparison to the mid latitudes, some field experiments took place on Kilimanjaro (Pepin et al., 2010; Appelhans et al., 2016) and on nearby Mount Kenya (e.g., Camberlin et al., 2014) and provide a useful regional framework. In addition to air temperature, the available measurements on Kersten Glacier at different elevations include air humidity and snow accumulation. The main goal of this paper is to provide a
documentation of the recorded variability in their gradients.

While previous studies related the observed gradients to local meteorological mechanisms, which revealed the importance of katabatic winds (Greuell and Böhm, 1998; Cartuan et al., 2015), a novel approach in the present study is to link the local observations in the glacier's surface layer to processes that act on the mesoscale of the mountain, with the help of a high-
resolution atmospheric model data set. The null hypothesis for our study is that (i) the vertical gradient in air temperature can be described by the canonical mean lapse rate of $-6.5$ K km$^{-1}$; (ii) relative humidity is constant with elevation; and (iii)

---

[1] A summary is available at http://thomasmoelg.info/factsheet_kili.pdf



snowfall amount decreases with elevation as expected in a convectively shaped precipitation environment at high altitude (Hastenrath, 1991).

## 2 Methods and Data

**2.1 Study region and measurement program**

The Kilimanjaro massif at the Kenya-Tanzania border is located close to the equator (~3°S) and shows remnants of a formerly larger glaciation in its central part, named Kibo. These small glaciers (surface area < 2 km² in 2011; Cullen et al., 2013) have been the attention of a collaborative project that has run for more than a decade, and thus a small network of automatic weather stations (AWSs) has been maintained since. While the published research topics span large-scale

atmosphere–ocean linkages to the glaciers, mesoscale circulation influences, and local mass and energy budgets, two AWSs were placed with the intent to gain knowledge of elevation gradients in the basic meteorological variables. These stations (AWS3 and 4) are situated on Kersten Glacier, the largest remaining slope glacier at present (Fig. 1). Its altitude range in 2011 (Cullen et al., 2013) was from almost the summit of Kibo (Uhuru Peak, 5895 m) down to ~5100 m. While the aerial photography shows the glacier extent in 2005 (Fig. 1), later visits confirmed that Kersten has split up in a region between

AWS3 and AWS4. This development had been expected due to the negative glacier mass balance and the shape of its elevation profile (Mölg et al., 2009a).

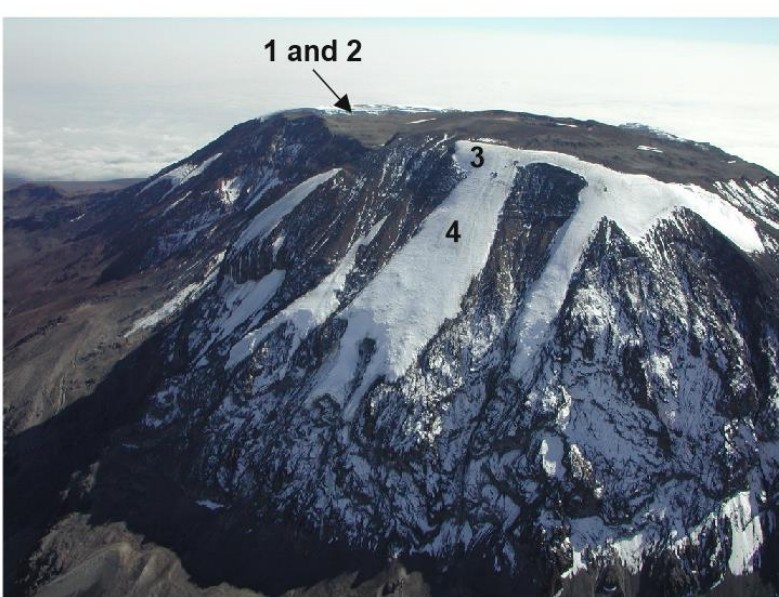

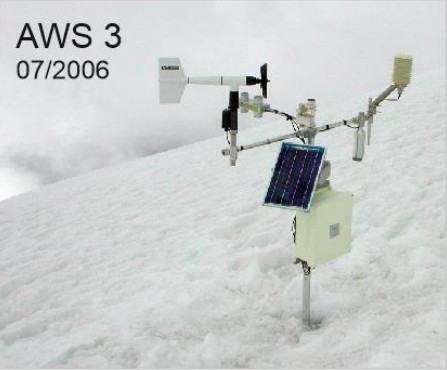

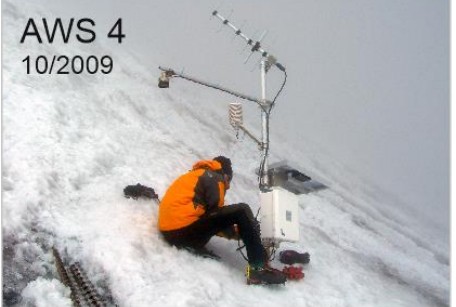

**Figure 1: Aerial photo of Kilimanjaro's central part Kibo in July 2005, indicating the locations of the automatic weather stations. The pictures on the right side show the data sources for this study in more detail (photos: N. J. Cullen, G. Kaser, R. Prinz).**



AWS3 was installed on 8 February 2005 at 5873 m and (like AWS4) is powered by a solar panel. It records air temperature (*T*) and relative humidity (RH) at two heights (we consider the upper sensor at initially 1.75 m), air pressure, wind speed and direction (at initially 1.85 m), radiation fluxes in the short and longwave spectrum, and surface height change (a proxy of snow accumulation or mass loss) in half-hourly steps. Numerous publications have presented this station and its data in the meantime; perhaps most relevant for an overview is Mölg et al. (2009a), who detail climatological signatures from the first

three years of measurements as well as the procedure of data post-processing to detect erroneous measurements. The only significant data gap so far exists for 2014, when the station started to tilt and ultimately fell. Since first signs of the tipping mast are obvious early in 2014, for example in the wind and radiation data that require a horizontal reference, we decided to not analyze the 2014 data any further. AWS3 was re-erected in January 2015, yet its data have not been retrieved since.

AWS 4 was installed on 7 October 2009 at 5603 m with the aim to measure elevation gradients on Kersten. Its installation required significant logistical preparation and physical expense, since the glacier can only be accessed from the top through Uhuru Peak and is very steep below its uppermost section. Therefore, AWS4 follows a "smart-stake" design that was spawned from research in the ablation zone of the Greenland ice sheet (Steffen et al., 2002), and is only equipped with a *T*/RH sensor (at initially 1.1 m) and an ultrasonic ranging sensor to also measure the surface height change. The instruments

– Rotronic MP100A (*T*/RH) and Campbell SR50a (surface height) – are the same as at AWS3. Their measurements have been transmitted every minute to a receiver station placed close to AWS3, where (apart from the high altitude) access is relatively easy. In the end AWS4 delivered suitable data until September 2013 for all three variables. While the sonic records are generally prone to noise and require special care (Sect. 2.3), surface height at AWS4 was also the first of the recorded variables to exhibit systematic problems (very high and frequent fluctuations) in October 2013, whereas the *T*/RH records

appear reliable at least through 2014.

## 2.2 Quality control and measurement height

Based on the above circumstances, our analysis covers the overlapping period with reliable data from 8 October 2009 to 30 September 2013 (*n* = 1454 days). The *T*/RH data were post-processed and quality-controlled as described in Mölg et al. (2009a). No signs of systematic problems are evident in these records, and the extent of missing data is minimal (< 0.1%)

like in the earlier data (Mölg et al., 2009a). Post-processing of the surface height data was trickier and is detailed in Sect. 2.3.

A noteworthy aspect of the measurements concerns the stations' platform, which is a mast drilled into the ice due to the significant slope (~18° at AWS3 and ~40° at AWS4). This platform design has proven its worth for on-glacier measurements (Winkler et al., 2011), yet the height of sensors above the surface varies in concert with mass accumulation and ablation on

the glacier: between 1.0 and 2.8 m at AWS3, and 0.8 to 1.9 m at AWS4 over the analysis period. Ideally, *T*/RH were available for the standard 2 m screening height. It is still unlikely that our non-constant measurement height affects the results seriously for several reasons. First, any longer intervals of contrasting surface height changes that would lead to

divergence of the thermal regimes at the stations (e.g., height increase at one station, decrease at the other) are absent over

the four-year period (Sect. 2.3). Secondly, mean $|\Delta T|$ between the upper sensor at AWS3 and the lower one (0.7 m below) is

only 0.31 K over the full record (2005–2013), and therefore on the order of the nominal accuracy of the Rotronic instrument

(0.2 K). Thirdly, methods to correct for a constant reference height require critical assumptions (e.g., Oerlemans and Klok,

2002) and would introduce further non-quantifiable uncertainties. And fourth, strong vertical $T$/RH variations on sloping

glacier surfaces in mountains usually develop in connection with a katabatic wind speed maximum a few meters above the

surface (Denby and Greuell, 2000; Sauter and Galos, 2016). Katabatic forcing is, however, assumed to be minimal for

Kersten due to its environmental settings, which will be detailed in, and substantiated by, the results (Sect. 3.2).

Hence the gradients examined in this study are with high confidence dominated by elevation and not by the

micrometeorological, vertical $T$/RH profile in the lowermost meters above the surface. They should still not be considered as

elevation gradients for screening height (2 m), but rather for "near-surface"; many other studies have used this terminology

as well if the measurement platform inhibits to maintain a strict reference at 2 m (e.g., González and Garreaud, 2017).

**2.3 Processing of the sonic ranger records**

The SR50a records from Kibo must be carefully post-processed, due to intervals of noisy data. In our experience this is a

general problem with measurements over glacier surfaces, although details are rarely documented in the literature. A few

report-type papers with an operational focus (e.g., Mair and Baumgartner, 2010), papers on measurement technology (Ryan

et al., 2008), and single research articles (Hardy et al., 2003) are some exceptions that discussed the topic. They also

concluded that signal and noise are typically blended in the sonic height records. Therefore, we detail our algorithm to clean

the data in the Supplement. It is necessary for such procedures to consider site-specific empirical thresholds, and in total our

algorithm uses four parameters. Although the sensitivity to the parameter choice is small, we consider the difference between

two contrasting parameter combinations as one source of total uncertainty (see Supplement).


The noise in the SR50a records can result from the specific ambient conditions; in particular the air temperature dependency

of the speed of sound as well as strong-wind and blowing-snow events (Hardy et al., 2003; Ryan et al., 2008). Transducer

degradation is another relevant factor. Further, there are adverse influences from a general perspective (listed in the

instrument manual), where two are inevitable for sloping glacier surfaces: (i) the sensor is not perpendicular to the target,

which is difficult on platforms where radiation and wind sensors require a horizontal reference surface; and (ii) an uneven

target surface. Both factors attenuate the return pulse of the instrument. Due to the noise we usually did not attempt to

interpret the hourly data but daily values at the most (e.g., Mölg and Hardy, 2004). This is also the case in the present work

for which we generated daily data (see Supplement) and calculated monthly and annual sums thereof.





Figure 2 illustrates that our algorithm succeeds in extracting the SR50a signal from the noise, and removes the "spikes" typical of erroneous measurements (Ryan et al., 2008). The record at AWS4 is less noisy for the 30-minute interval, since the one-minute logging at AWS4 allows us to detect the most obvious errors beforehand (see Supplement); at AWS3 the shortest logging interval is 30 minutes based on one-minute samples, since the data logger must store many other variables as well (Sect. 2.1). The sonic records demonstrate that the net surface lowering over the analysis period was stronger at the higher

altitude. This empirical finding agrees with mass balance model calculations showing a higher absorption of solar radiation at the glacier top and, therefore, a more negative local mass budget (Mölg et al., 2009a). The measurements also show that major intervals of snow accumulation coincide in both records (see letters a-g in Fig. 2). Our correction algorithm also enables days with "no data", if days are very noisy and lack any signal (see Supplement). If a month contains more than a week of missing days, we do not analyze this month further in the present paper ($n = 7$ months at AWS3).

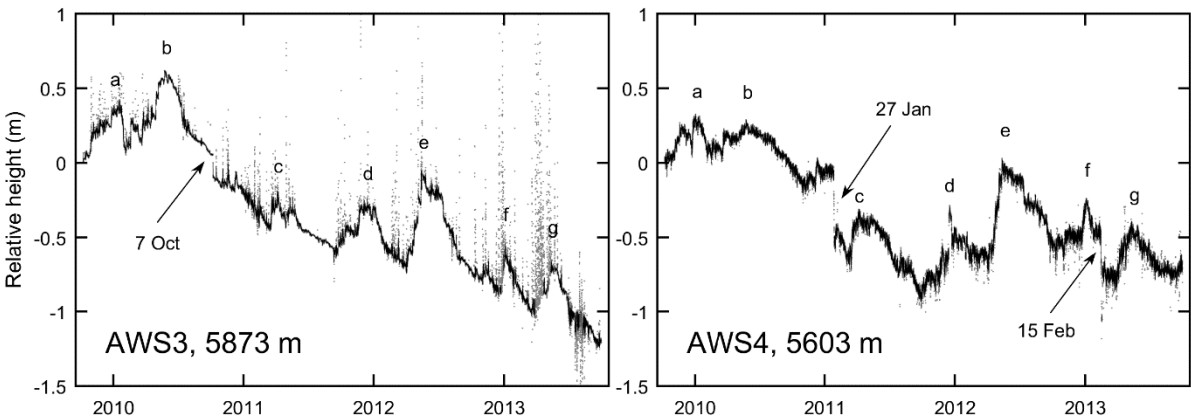


**Figure 2: Surface height change at AWS3 and AWS4 relative to 8 October 2009. Grey dots are the initial half-hourly data, which are less noisy at AWS4 due to the logging interval (see text). The black line illustrates corrected data (Parameter Combination 1, see Supplement). Entered dates indicate spurious "jumps" (see text), and letters a-g the main and consistent accumulation periods. Ticks on the x-axes mark the beginning of the respective year (1 January).**


It is worth mentioning that single, strong surface decreases occur on the scale of only a day (date entries in Fig. 2). While the "jump" on 7 October 2010 at AWS3 is due to field work (sensor change from SR50 to the successor model SR50a), the two jumps at AWS4 in early 2011 and early 2013 are harder to resolve. A potential "true" signal from atmospheric influences could be the sudden destruction of penitentes, an ablation feature that has been observed preferably on the southern slope

glaciers (see also early literature, e.g. Geilinger, 1936). A sudden increase in RH at AWS4 (from ca. 20% to 95%) in the second case would lend some support to this interpretation, since the destruction requires the impact of a moist air mass. More likely though are the two jumps due to melt-driven mast rotation, which is recognizable to some extent on our photos. Lowering of the cross-arm, on the other hand, can be ruled out as it would result in an (artificial) accumulation signal. In any case, the jumps imply ablation which is not analyzed in this study.



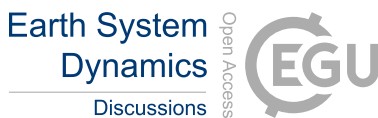

The second area of data processing after the "clean" signal extraction (above) concerns the efforts to (i) compare the accumulation amounts at AWS3 and AWS4 for the elevation gradient determination, and (ii) compare accumulation amounts with conventionally measured precipitation at lower elevations on the mountain. The Supplement also details these two procedures. In short, point (i) arises from the different slope at the measurement sites, which impacts the sonic signal due to the measurement cone principle. Point (ii) involves the conversion of (solid) accumulation to (liquid) water equivalent (w.e.)

amounts by assuming a fresh snow density. Both steps introduce further uncertainties and contribute to the total uncertainty (see Supplement) that is reported hereafter for the sonic ranger-derived accumulation.

**2.4 Atmospheric model experiments**

A high-resolution data set from a decadal-scale atmospheric model run (2005–2017) is available from a recent investigation looking at large-scale climate variability effects on Kilimanjaro (Collier et al., 2018). The model grid of that study employed

a horizontal resolution of 800 m over the entire Kilimanjaro region, resulting in a model summit of 5827 m a.s.l. and a good representation of the glacierized elevations (Fig. 3). We emphasize that our goal here is not to evaluate this atmospheric model run further. Collier et al. (2018) made the evaluation and showed that the model reproduces the important features of atmospheric conditions in the summit zone. We thus assume that the mesoscale processes over the mountain, which are presented in the results section, also contain a reasonable degree of realism. Beyond that, Kersten Glacier is represented by

only one grid cell in the model despite the high resolution; evaluating simulated gradients on the glacier scale would therefore not be possible.

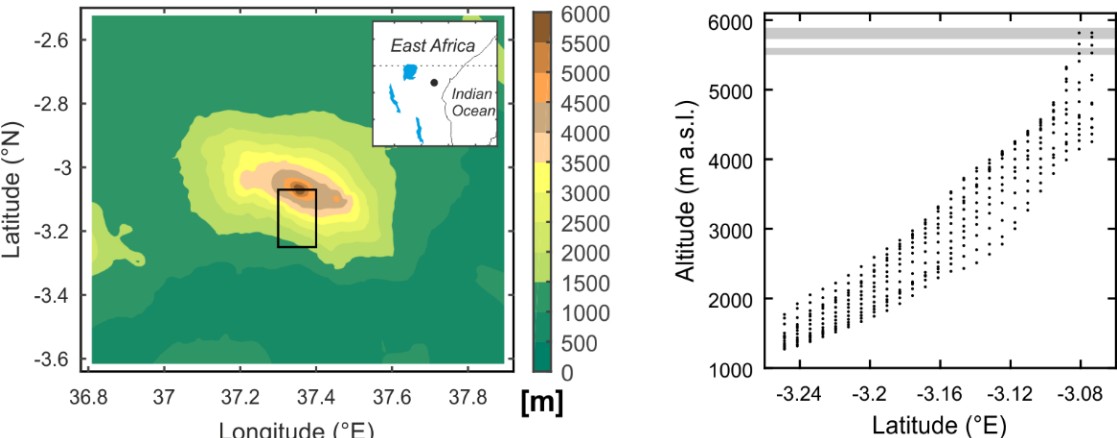

**Figure 3: (Left) Topography of the innermost domain in the atmospheric modeling study of Collier et al. (2018). The black box**
**delineates the "southern mountain slope" defined in the present study. Inset in the upper right corner illustrates the full area**
**covered by the model's (outermost) computational domain and the location of Kilimanjaro (dot). (Right) Meridional profile of**
**altitude on the southern slope indicated by the single grid points; the defined upper and lower summit zone are highlighted by**
**grey bars.**



Two earlier Kilimanjaro studies (Mölg and Kaser, 2011; Mölg et al., 2012) provided a guideline for the aforementioned work. They employed the same atmospheric model in a very similar grid setup and presented simulations for August 2005 and April 2006, which are a typical month of the dry and wet season, respectively. We will also refer to the results of these studies with regard to mesoscale processes that affect the mountain.

Given the location of Kersten Glacier, we define the "southern (mountain) slope" as the area of the black rectangle in Fig. 3. In this area we distinguish between the "upper summit zone" (grid points > 5720 m) and the "lower summit zone" (grid points in the 5500–5600 m band) for calculating meteorological gradients in the model data. The difference between the mean altitudes of the two zones is 272 m, which is almost identical to the AWS3/AWS4 difference (270 m). Again, Kersten Glacier is not resolved by multiple grid points in the model, hence the comparison of model gradients with observations will

shed light on how much the general gradients in Kibo's summit zone (as represented by the model) resemble the ones in the local glacier environment (as captured by measurements).

From the model output we calculated heat advection over the mountain surface with the potential temperature ($\theta$) as $\partial\theta/\partial t = -u\ \partial\theta/\partial x - v\ \partial\theta/\partial y$, where the horizontal zonal ($u$) and meridional wind ($v$) are a good proxy of cross-slope and along-slope

transport, respectively, for our environment. We repeated all wind-based analyses with the horizontal wind vector projected onto the terrain vector, yet the results differed marginally. Thus we only present the results from horizontal winds, in order to maintain consistency for the comparison with observed (horizontal) winds. As the focus of the paper is the observations, we restrict all analyses to "near-surface" (model output for the nominal 2 m height or, in case of winds, 10 m), vertically integrated metrics, or output for the first model level (~16 m above the surface).

**3 Results and discussion**

In the following we report all elevation gradients per 100 m, which seems practical with regard to the altitude extent of most present mountain glaciers (typically less than several km) and distinguishes from micrometeorological gradients (discussed in Sect. 2.2) that are usually given per meter (e.g., Greuell and Böhm, 1998). The accumulation signal from the sonic ranger may also capture snow redistribution during wind drift, but its importance on Kibo slopes is limited in our experience. The

derived quantity should thus be a good proxy of solid precipitation, yet we will continue to call it accumulation (ACC) to not forget that the measurement principle differs from a regular precipitation gauge.

**3.1 General climatic situation**

Although the climatic characteristics of our Kilimanjaro records were repeatedly presented in previous studies (e.g., Mölg et al., 2009a; Kaser et al., 2010; Hardy, 2011) and were also set into a wider context of meteorological records from tropical

glaciers (Nicholson et al., 2013), a short summary for the analysis period will help to get a feel for Kersten's environment. This summary puts emphasis on the previously unpublished AWS4 data. Figure 4a illustrates the key characteristic of the





regional climate, which is a succession of dry and wet seasons in the course of the year. ACC and RH at AWS4 typically peak around April and again late in the year; such empirical patterns gave rise to the frequently used distinction between the "long rains" (March–May, MAM) and "short rains" (October–December, OND) season of equatorial East Africa. A

pronounced dry season from June to September (JJAS) and a short and dry (yet more variable) season in January and February (JF) complement the cycle. The error in the ACC estimation is mostly small, and the cases with a somewhat pronounced uncertainty (max. ± 18 mm w.e. per month) do not put the measured pattern in question. The bimodal distribution of precipitation also applies to Kilimanjaro and Mount Kenya at the mountain scale (Camberlin et al., 2014; Appelhans et al., 2016).


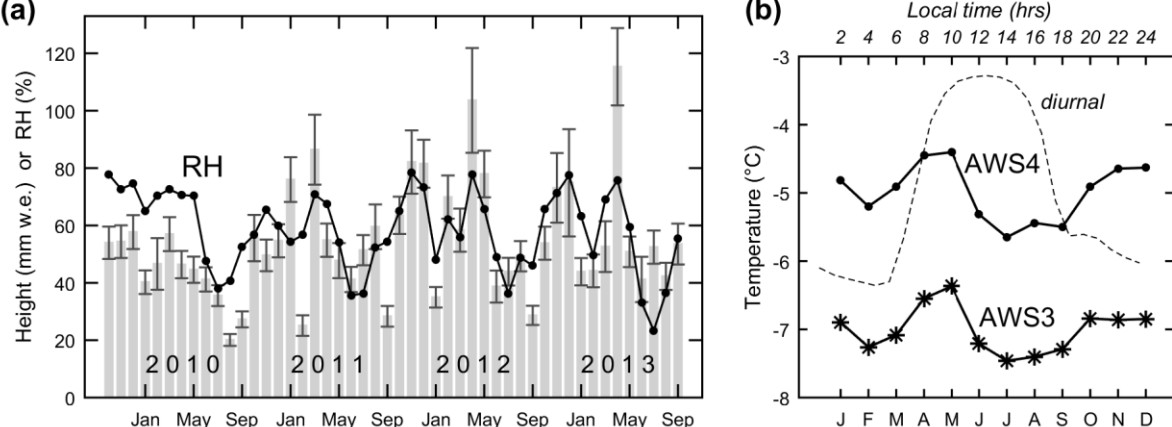

**Figure 4: (a) Monthly accumulation totals (bars) and mean monthly relative humidity (line) at AWS4, and (b) mean annual cycle of air temperature at both AWSs and the associated mean diurnal cycle at AWS4 (dashed line and upper x-axis) between 8 October 2009 and 30 September 2013. Error bars in (a) are from the sonic ranger data processing (Sect. 2.3).**


The monthly means of air temperature, by contrast, show little variation at both stations (Fig. 4b), which also applies to the East African lowlands and to the tropics in general. Stronger and systematic variability in $T$ only concerns the diurnal cycle, as illustrated in Fig. 4b. The weak seasonality of $T$ also implies that the seasonal RH cycle (Fig. 4a) is mostly driven by the variability in water vapor content of the air masses. Table 1 puts the temperature and moisture conditions in a wider climatic

context, with further data from AWS3. One can notice a very dry and cold, solar radiation-rich environment with high global radiation yet low incoming longwave radiation and water vapor pressures. Mean wind speed is moderate, and the hourly values show only a low percentage of winds > 10 m s$^{-1}$ (2.1 %). Table 1 also suggests that elevation gradients exist between AWS4 and AWS3, noticeably for $T$ (see also Fig. 4b), ACC, and water vapor pressure. RH, on the other hand, hardly differs. The moister conditions at AWS4 can be anticipated from field observations, since AWS4 elevations are enveloped by clouds

more frequently than the regions at and above the crater rim (Kaser and Osmaston, 2002); the associated physical basis (Mölg et al., 2009b) is detailed further in the remainder of Sect. 3.



**Table 1: Measured mean values at AWS3 and AWS4 between 8 October 2009 and 30 September 2013. The unmeasured variables at AWS4 are denoted NA.**

|  | AWS3 (5873 m) | AWS4 (5603 m) |
|---|---|---|
| Global radiation (W m$^{-2}$) | 341 | NA |
| Incoming longwave radiation (W m$^{-2}$) | 180 | NA |
| Wind speed (m s$^{-1}$) | 4.1 | NA |
| Air temperature (°C) | −7.0 | −5.0 |
| Relative humidity (%) | 57% | 58% |
| Water vapor pressure (hPa) | 1.98 | 2.40 |
| Air pressure (hPa) | 502 | NA |
| Accumulation (mm w.e. year$^{-1}$) | 334 ± 35 | 643 ± 76 |


## 3.2 Air temperature gradient

The mean $T$ gradient on the glacier over the analysis period amounts to –0.75 K per 100 m. This gradient is stronger than the universal mean lapse rate of –0.65 K per 100 m and the one in the non-glaciated high elevations between 4000 and 5000 m on Kibo's southern slope (Duane et al., 2008; Appelhans et al., 2016). Both make sense in light of the overly dry climatic

environment of the summit zone. To our knowledge no measured $T$ gradients on tropical glaciers have been published before, but data from extratropical mountain glaciers measured over more than one season (see table 1 in Petersen and Pellicciotti, 2011) also show slightly weaker gradients.

### 3.2.1 Seasonal and diurnal variabilities

The annual cycle demonstrates variability around the mean value (Fig. 5a). The most pronounced variability and the extreme

values occur in the first 13 months of the record, which is most likely related to large-scale climate variability affecting the summit zone – El Niño around the 2009/2010 transition and a switch to La Niña and a negative Indian Ocean Dipole event around October 2010 (Collier et al., 2018). Over the full record, however, a pattern emerges with stronger gradients in the wet than in the dry season, e.g., local negative peaks mostly occur in MAM or OND. This observation seems in conflict with theoretical considerations, where one would expect deviations from the mean towards the dry-adiabatic lapse rate in dry

conditions and towards the moist-adiabatic lapse rate in wet months. Such expected variabilities were indeed measured in mountains for larger spatial units, for example at the valley-scale in High Asia over the wet (monsoon) season and the preceding/subsequent dry seasons (Shea et al., 2015). However, these valley lapse rates predominantly apply to "ambient" conditions, from which the temperature regime in the glacier surface layer differs (Greuell and Böhm, 1998). Besides the microclimatic factor, regional temperature seasonality requires attention. Although it is small at the equator, its amplitude

increases as one moves downslope on the mountain and, hence, stronger (weaker) $T$ gradients along Kilimanjaro's southern slopes tend to occur during the slightly warmer (colder) season (Appelhans et al., 2016). Consider this effect for Kersten





Glacier, where the seasonal fluctuation around the mean $T$ is also slightly larger at AWS4 (i.e. the lower, warmer site) than at AWS3 (Fig. 4b). Relatively large negative anomalies at AWS4 in JJAS, for instance, will therefore contribute to weakening the $T$ gradient in this (dry) season (Fig. 5a). On top of the "background" temperature influence, the specific mesoscale

atmospheric circulation of a mountain must be accounted for. In this respect (i) our high-resolution atmospheric modeling for Kilimanjaro has produced the consistent result of weaker $T$ gradients above 5000 m in the dry season (e.g., table 2 in Mölg and Kaser, 2011), and (ii) the mean diurnal cycle of the $T$ gradient (see below) warrants attention and helps to understand the seasonal variability further.

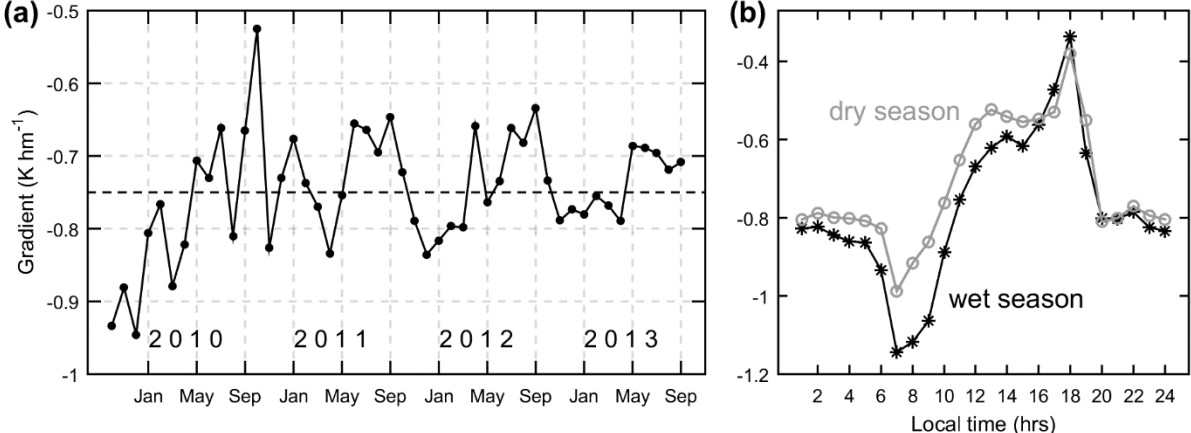


**Figure 5: (a) Mean monthly $T$ gradient (dotted line indicates the four-year mean) and (b) its mean diurnal cycle for the climatological dry (JF, JJAS) and wet (MAM, OND) seasons in gray and black, respectively, between 8 October 2009 and 30 September 2013. Note the different y axis scaling.**

This diurnal cycle (Fig. 5b) demonstrates that the stronger gradients during the wet season result from a development that begins around midnight and produces the maximum difference in the early morning hours around 8:00 local time (LT), after which the difference decays and vanishes from 16:00 LT onwards. This decay is linked to another key feature of the mean diurnal cycle regardless of season, namely the occurrence of the weakest gradients in the afternoon (Fig. 5b). In this respect the role of katabatic winds over glaciers was emphasized in theoretical (Greuell and Böhm, 1998) and empirical work

(Petersen and Pellicciotti, 2011), which underlined that reduced $T$ gradients coincide with the daily peak of the katabatic winds. Katabatic forcing is, however, assumed to be minimal for Kersten since the glacier (i) is very small; (ii) has no surrounding topography that promotes katabatic winds (e.g., valley slopes); (iii) is therefore, and due to its high altitude as well, strongly exposed to the geostrophic wind of the mid troposphere; and (iv) lacks the basis for strong horizontal pressure gradient forces as the temperature difference between glacier surface layer and ambient air remains small (Pepin et al.,

2010), unlike on mid-latitude glaciers where this difference leads to a well-developed local katabatic layer (Greuell and Böhm, 1998). The more likely drivers of the observed diurnal cycles on Kersten (Fig. 5b) must therefore be sought in the



interplay of local with regional and larger-scale factors – a perspective that will also be relevant for the humidity gradients (Sect. 3.3). In particular, intensified daytime transport and mixing by the large-scale geostrophic wind or the regional flow over the mountain (i.e. the *non-local factors*), and elevation-driven variability in the energy balance (i.e. the *local factors*)

must be considered hereafter.

### 3.2.2 Non-local control factors

Turning to the first viewpoint, the atmospheric model's diurnal cycle of the $T$ gradient in the summit zone of the southern mountain slope (Fig. S3) shows the same basic variability as the measurements on the glacier (Fig. 5b), which suggests a

link of the observed cycle to the mesoscale circulation. Its key role was already underlined by Troll and Wien (1949) and corroborated in many studies in tropical mountain meteorology and climatology (e.g., Mendonca, 1969; Mölg et al., 2003; Wang and Kirshbaum, 2015). The role is based on the year-round strong solar irradiance in this climate, which initiates upslope winds in the early daytime hours and intensifies them in the course of the afternoon, leading to convective cloud formation in the summit zone. This process can be anticipated on Kibo by looking at the mean diurnal cycle of (horizontal)

wind speed measured at AWS3 (Fig. 6, top). The nighttime "geostrophic" value of ~4.4 m s$^{-1}$ weakens during the day, indicating increasing influence of the mountain-forced vertical winds and the expected stronger signature in the wet season (due to stronger convection). The atmospheric model data underline that there is a clear switch between daytime ascent, and nighttime descent, of the air mass over the southern mountain slope (Fig. 6, middle), which exhibits the classical concept of the diurnal circulation over tropical mountains from the numerical modeling angle. Furthermore, the daytime upslope

movement is obviously linked to an increase in convection that is stronger in the wet season (Fig. 6, bottom).




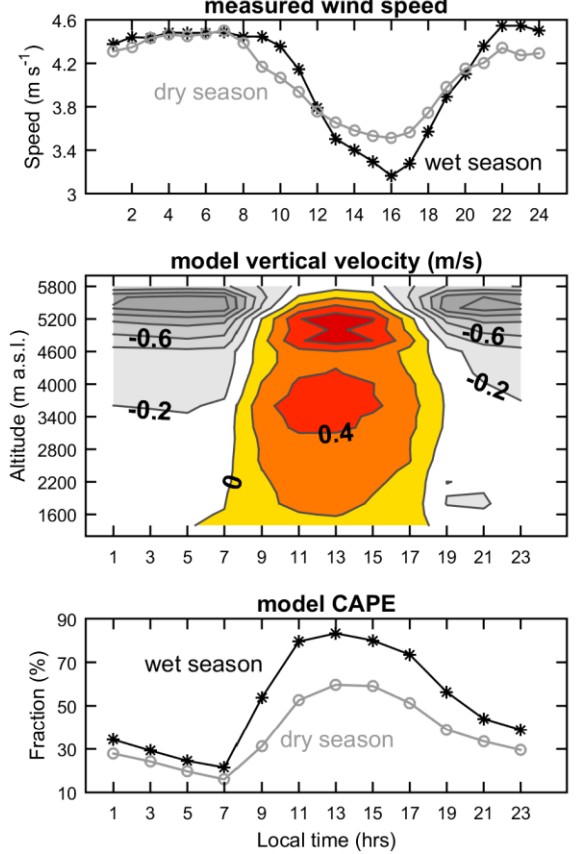

**Figure 6: Mean diurnal cycles of (top) measured wind speed at AWS3, (middle) vertical velocity on the first model level (17 m above ground; contours use 0.2 m s⁻¹ equidistance) over Kibo's southern slope versus atitude, and (bottom) modeled areal fraction of convective available potential energy (CAPE) > 0 J kg⁻¹ on Kibo's southern slope between 8 October 2009 and 30 September 2013. Top and bottom figures distinguish between the climatological dry (JF, JJAS) and wet (MAM, OND) seasons in gray and black, respectively. The x-ticks in the model plots indicate the available hours of model output.**

A logical consequence of this circulation constraint with regard to the weakening $T$ gradients between sunrise and sunset (Fig. 5b) would be increasing upslope warm-air advection as the day progresses, driven by the mesoscale circulation rather than the large-scale geostrophic wind. An analysis of potential temperature advection over the model's southern mountain slope (Fig. 7a) helps to challenge this notion. The bottom part of Fig. 7a showcases three interesting features in this regard, using the dry season (the same features exist in the wet-season data). First, there is a sharp transition between advective cooling and warming as night and day alternate, respectively. This pattern coincides with the air mass descent during night and the ascent during daytime, as discussed above. Secondly, the amplitude of this cooling/warming pattern increases with increasing altitude. And thirdly, the maximum rate of advective heating is shifted back in daytime going from low- to high-elevation zones. It occurs as early as 9:00 LT at 2000 m, and not before 13:00 LT at 4000 m and higher. These features together indeed suggest that the mesoscale circulation communicates the heating signal from the lowlands to higher elevations over the course of the day and mixes potentially warm air upslope; the intensification of this process with altitude helps to explain why daytime elevation $T$ gradients in the summit zone are reduced (Figs. 5b and S2). At the same time, the magnitude of the local vertical turbulent sensible (QS) and latent (QL) heat fluxes increases in the model's upper summit zone ($|\overline{QS + QL}|$ = 228 W m⁻² from 9:00–17:00 LT, compared to 16 W m⁻² for the other hours of day), which is known to

be favored by warm-air advection (e.g., Kang et al., 2014). This daytime increase was also found for the measurement-based determination of turbulence on the summit glacier surfaces (Mölg and Hardy, 2004).

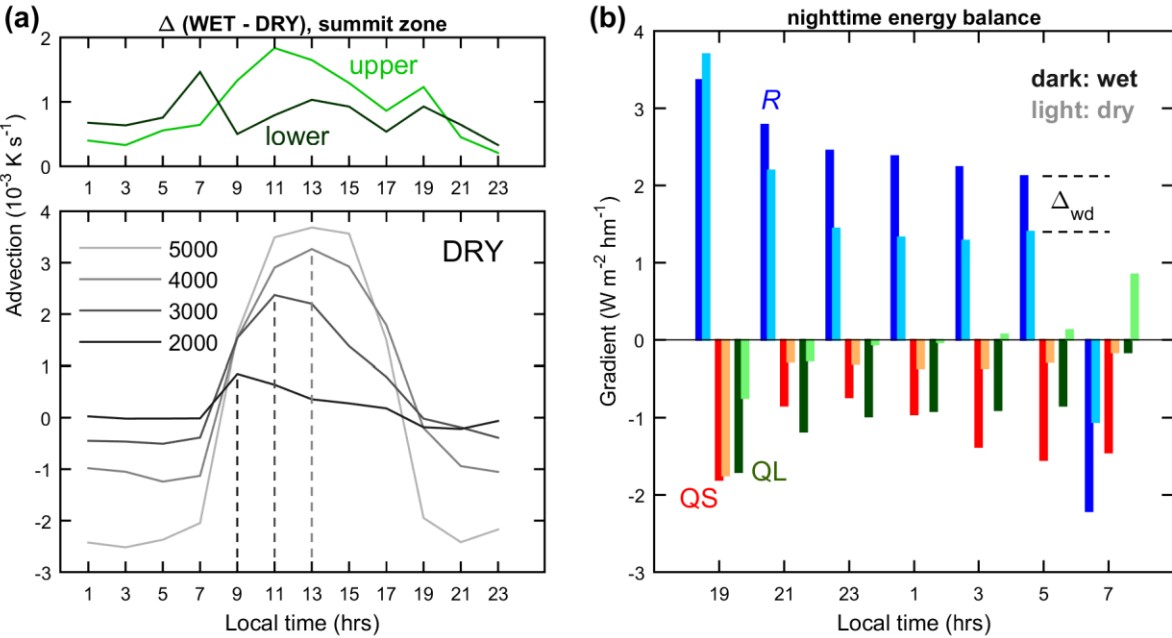

Figure 7: (a) Mean diurnal cycle of potential temperature advection near the surface of the southern slope (bottom) in the dry season for different elevation bands (± 100 m from the given center altitude; vertical lines indicate the maxima between 2000 and 4000 m) and (top) of the related differences in the wet season for the upper (> 5716 m) and lower (5500−5600 m) summit zone. (b) Mean elevation gradients in the climatological dry (JF, JJAS) and wet (MAM, OND) seasons in light and dark tones, respectively, for the summit zone in the course of the night: sensible (QS) and latent (QL) turbulent heat flux and net radiation (R). The gradient difference between wet and dry season ($\Delta_{wd}$; discussed in Sect. 3.2.3) is exemplified for R at 5:00 LT. Plots are based on the atmospheric model output for the period 8 October 2009 to 30 September 2013.

The top plot of Fig. 7a singles out the model's summit zone to elaborate the seasonal differences in the T gradients further. Again, stronger gradients in the wet season (Fig. 5) either result from the upper summit zone with AWS3 becoming colder or the lower summit zone with AWS4 becoming warmer (or a combination of both), relative to the dry season. In Fig. 7a, top, the "lower zone relatively warmer" case appears, since advective cooling in the wet season is attenuated more strongly in the lower than in the upper summit zone from 21:00 LT onwards (i.e. $\Delta$ is more positive in the lower zone), a tendency that is maintained for rest of the night and peaks at 7:00 LT. Around this time, the consistent characteristic of stronger wet-season T gradients is most pronounced both in the observed (Fig. 5a) and modeled (Fig. S3) diurnal cycles. However, the quantitatively more prominent feature concerns daytime (Fig. 7a, top), when warm-air advection during the wet season is clearly higher in the upper summit zone. This is another process consistent with the observed (Fig. 5b) and modeled (Fig. S3) gradients, in this case the daytime decay of the difference between wet- and dry-season T gradients. The mesoscale

circulation-driven heat advection, therefore, appears to be a key mechanism for the diurnal variation of $T$ gradients and their seasonal variabilities, particularly during daytime. The importance of the mesoscale circulation also appeared for Mount
Kenya in a qualitative discussion of glacier modeling results (Prinz et al., 2016).

### 3.2.3 Embedded local factors

Nonetheless, the smaller seasonal differences in advection during night in the summit zone also suggest to revisit the energy balance viewpoint expressed above. Figure 7b puts together the local elevation gradients in the modeled QS, QL, and net radiation ($R$) for the summit zone during night. The $R$ gradient is positive with altitude, which stems from less longwave
cooling in the upper summit zone and implies a contribution to reducing $T$ gradients during night. Since such a reduction is neither evident in observations nor model, stronger and opposing effects must exist. The upslope increase in advective cooling during night is one of them (Fig. 7a), but the local turbulent fluxes also indicate enhanced cooling with elevation by their prevailing negative gradients (Fig. 7b). Regarding the seasonality of $T$ gradients, the difference between the wet- and dry-season ($\Delta_{wd}$) gradients in Fig. 7b is decisive. Since $\Delta(\partial QS/\partial z)_{wd} + \Delta(\partial QL/\partial z)_{wd}$ in the core of the night (21:00 to 5:00
LT) is more negative than $\Delta(\partial R/\partial z)_{wd}$ is positive, the net effect of the local energy balance in the wet season is to make $T$ gradients stronger during night, and even more so at 7:00 LT when $\Delta(\partial R/\partial z)_{wd}$ is also negative. This effect is consistent with measured and modeled cycles of $T$ gradients (see above). Such a consistency does not exist for the daytime interval after 7:00 LT; thus, at least for the night and early morning the results suggest that the local energy balance and temperature advection together shape the diurnal cycle of $T$ gradients.

## 3.3 Humidity gradient

The mean RH gradient is close to zero, which was already obvious from Table 1. Duane et al. (2008) also found almost constant RH in their measurements at 5500 m (over bare rock) and 5800 m (over glacier). Below ~5000 m, RH seems to start increasing towards the maximum that is reached in the rain forest belt around 2400 m (Appelhans et al., 2016). Thus it appears that the absence of a mean RH gradient is typical at least for the uppermost 400 m of the mountain, and not
necessarily unique to the air over Kersten Glacier. Again, we are not aware of any published RH gradients on tropical glaciers, but RH measurements on mid-latitude glaciers yielded a near-constant RH for altitudinal extents of a few hundreds of meters too (e.g., table 2 in Greuell et al., 1997).

### 3.3.1 Seasonal and diurnal variabilities

The seasonal and the mean diurnal variabilities in the elevation gradient of RH (not shown) are also small: monthly means
settle almost exclusively in the ±2 % range per 100 m, while the diurnal cycle shows nighttime gradients close to zero and a peak strength of only −5 % per 100 m at noon. Since RH is driven by $T$ and the water vapor content of the air, the pronounced variability in $T$ gradients (Sect. 3.2) implies that there must be some co-variability in the gradients of water



vapor content. Hence, we focus on water vapor pressure rather than on RH in the rest of Sect. 3.3, which also provides more insight into governing processes.


Figure 8a illustrates the seasonal variability of the vapor pressure gradient and reveals a pattern with small gradients mostly in June and July, and the strongest gradients in January and February. This means that the largest contrasts occur between the establishment of the long dry season in boreal summer and the transitional (mostly but not always dry) season in boreal winter. Observations of potential cloud frequencies are a likely explanation of this pattern. Pepin et al. (2010), for example,

suggested that the greatest difference in cloud formation cycles on Kilimanjaro exists between the two dry seasons – clouds in January regularly manage to ascend to the crater rim, while the atmosphere in July is generally too dry for significant cloud formation at high altitude. AWS4 data also show that the difference in frequency of near-saturated air (hourly RH > 95 %) between MAM and OND is a mere 5 % units, but it is on average 12 % units higher in JF than in JJAS with an hourly peak of +29 % units in the 13:00–14:00 LT window.


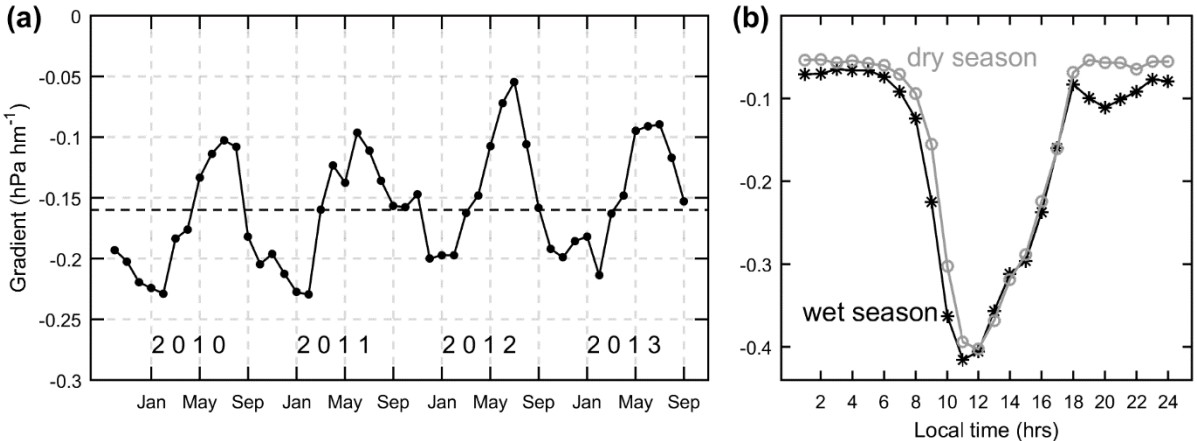

**Figure 8: (a) Mean monthly water vapor pressure gradient (dotted line indicates the four-year mean) and (b) its mean diurnal cycle for the climatological dry (JF, JJAS) and wet (MAM, OND) seasons in gray and black, respectively, between 8 October 2009 and 30 September 2013. Note the different y axis scaling.**


Differences in the mean diurnal cycle of the vapor pressure gradient between the classical East African wet and dry seasons (Fig. 8b) are therefore less obvious than for the $T$ gradient (Fig. 5b). However, the general variability of the vapor pressure gradient over the course of a day shows a prominent pattern (Fig. 8b): weak gradients prevail during the night, while the morning sees a strengthening of the upslope decrease that produces a sharp negative peak around noon. The return to the

nighttime weak gradients is less rapid and is accomplished by early evening. A general co-variability with the $T$ gradients is therefore evident at the diurnal scale, but most interestingly the negative peak occurs some hours delayed compared to the





negative peak in the *T* gradients (Fig. 5b). This timing shift produces the small, yet existing RH gradient of –5 % per 100 m at noon mentioned at the start of the section.

### 3.3.2 Non-local control factors

The local data and inferences above suggest that, in light of the ascending air masses during daytime, significant water vapor transport must occur in the summit zone. At this point we need to step to the mesoscale again, for which the previous section on *T* gradients (Sect. 3.2) has created a useful basis. The step is warranted as the atmospheric model's diurnal cycle of the vapor pressure gradient in the summit zone (Fig. S4) shows again a similar variability to the on-glacier measurements (Fig. 8b).


Figure 9 combines a vapor pressure and wind analysis to demonstrate model patterns of the vapor transport mechanism on the mountain. Starting in the middle of the night (upper left corner), the northerly component in the horizontal wind vector indicates a well-developed downslope flow in both seasons persisting until 7:00 LT. Only below ~3000 m do some vectors indicate a stronger east-west component, probably an effect of flow deflection at low elevations as studied in detail by Mölg

et al. (2009b) and discussed later in Sect. 3.4. The water vapor changes in the 1:00 to 5:00 LT window show drying at all altitudes above 2000 m. Only at 7:00 LT and in the wet season, first signs of moistening appear in the lower half of the mountain. As insolation increases rapidly in the morning, the switch from drying to moistening appears prominently at 9:00 LT and maximizes in the 2000 to 3500 m altitude zone that contains the rainforest belt, a major source of local moisture (e.g., Hemp et al., 2006). At the same time, the daytime upslope circulation starts to establish from the mountain base

upward and turns the horizontal wind to southerly. In the subsequent hours including 11:00 and 13:00 LT, however, the largest water vapor increases occur above the forest belt. The moisture excess from low-levels, therefore, is transported upslope in that time window. After 15:00 LT the mountain enters a regime of decreasing water vapor at almost all altitudes, while the transition to the downslope circulation (with northerly winds) occurs between 17:00 and 19:00 LT when solar insolation ceases. The succession of a moistening and drying phase in the course of the day was also suggested by Pepin et

al. (2010) from the comparison of along-slope measurements with reanalysis data, the latter of which mostly reflect larger-scale air mass properties without mountain climate influence.



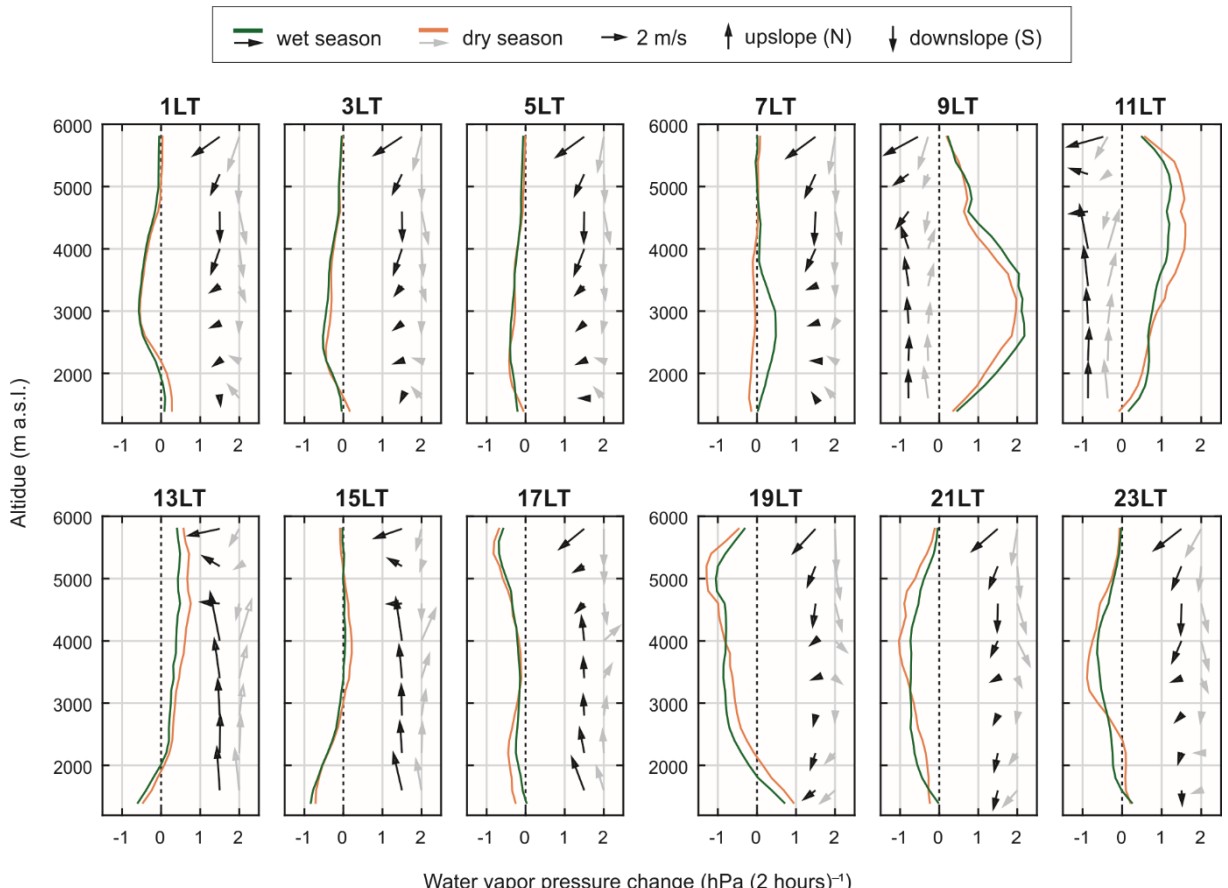

**Figure 9: Mean diurnal cycle of the 2-hourly water vapor pressure change (until the indicated time) on Kibo's southern slope at**
**2 m height for the climatological dry (JF, JJAS) and wet (MAM, OND) seasons in brown and green, respectively, between 8**
**October 2009 and 30 September 2013. Area-averaged mean horizontal wind vectors at 10 m height are drawn for 600 m bins from**
**1600 m to 5800 m a.s.l. (as on a map from north) at arbitrary $x$ axis locations, typically at $x = 1.5$ (wet season, black) and $x = 2$ (dry**
**season, grey) except for 9:00 and 11:00 LT to avoid overlap with the line plot. Note that the change analyses obfuscate the fact that**
**the bulk of the water vapor (mean pressure > 10 hPa day-long) is always found between the mountain base and 2500 m.**


A further point of interest here is the result that the daytime upslope circulation in the uppermost 1000 m seems to be more

effective in the wet season; occasionally, winds indicate downslope flow in the dry season when the wet-season counterpart

shows an upslope flow (e.g., 11:00–15:00 LT in Fig. 9). This tendency is in line with the stronger daytime temperature

advection in the summit zone during the wet season (Fig. 7a). The point raised is furthermore tied to the stronger daytime

upslope circulation than nighttime downward flow in the model (Fig. 9), which is a key feature of the thermally driven

circulation in mountain systems (e.g., Mendonca, 1969; Giovannini et al., 2017). Only the uppermost wind vectors drawn in

Fig. 9 seem less affected by the mesoscale circulation and show a rather constant flow from northeasterly directions (mostly

20-80°). This is the typical large-scale horizontal wind at 500 hPa over Kibo (Mölg et al., 2009b), which is reflected in the

measured horizontal wind direction at AWS3 (43% of the hourly distribution settle in the 20°-80° sector) and reaffirms





earlier assessments that the very highest elevations on the mountain are impacted vitally by the free tropospheric flow (Mölg et al., 2009a, 2009b).

### 3.3.3 Embedded local factors

Regarding gradients on Kersten, an interesting feature occurs above 5500 m in the moistening phase from after 7:00 LT to noon (Fig. 9); rates are weaker at the summit than at the elevations beneath. The measured vapor pressure gradient cycle
with intensifying elevation gradients before noon (Fig. 8b) is consistent with this result. Also consistent with measurements is the drying phase between 17:00 and 19:00 LT with its again weaker rates at the summit than beneath, which attenuate the vapor pressure gradients towards the evening. However, obvious daytime weakening of vapor pressure gradients in the model only happens after 15:00 LT (Fig. S4), while the weakening in the measurements (Fig. 8b) already starts in the early afternoon. A possible explanation comes from glacier mass balance modeling for the summit ice fields, which showed that
sublimation amounts at AWS3 are on average ~20% higher than at the AWS4 altitude (figure 9 in Mölg et al., 2009a). Since sublimation peaks around noon and in the early afternoon (Mölg and Hardy, 2004), a characteristic also found on other dry and sublimation-rich glaciers (e.g., MacDonnel et al., 2013), the potentially largest influence on the local water vapor gradients fits with the time window of interest. Support for the sublimation amount impact is the fact that the measured water vapor saturation deficit differs most clearly in the early afternoon, when the air at AWS4 shows a weaker reduction of the
deficit, and switches into the mode of saturation deficit increase sooner, than the air at AWS3 (Fig. S5). Hence it is likely that a local mechanism not resolved in the mesoscale atmospheric model (differential ablation) impacts the water vapor gradients during the early afternoon. With the arrival of night, more consistency of local measurements and the mesoscale patterns emerges again, manifested as the establishment of near-constant water vapor changes above 5500 m from 21:00 LT onward (Fig. 9) and the measured near-constant nighttime gradients (Fig. 8b).

## 3.4 Accumulation gradient and precipitation

The mean amount of measured ACC on Kersten Glacier clearly decreases with altitude, which Table 1 pointed out. Considering the uncertainty in that table, the mean annual gradient over our four-year analysis period is −114 ± 16 mm w.e. per 100 m, or −18 ± 2.5 % per 100 m in relative numbers (Table S1). The sign of the measured gradient fits well with the general knowledge of tropical mountain climate; observations around the globe have indicated that precipitation maximizes
at mid-mountain elevations, where the available precipitable water peaks in a predominantly convective atmospheric environment, and decreases above toward the summit (e.g., Hastenrath, 1991; Anders and Nesbitt, 2015). Again, we are not aware of any published precipitation gradients on tropical glaciers. Yet it is safe to say that average gradients on mid- and high-latitude glaciers differ substantially from the tropical case, since the different nature of precipitation formation in the extratropics (with a stronger role of frontal and stratiform precipitation) produces increasing mean amounts towards the
summit (e.g., Rögnvaldsson et al., 2007). The effect of precipitation formation type on the elevation of maximum amounts





can even be seen within one region, if seasons with predominantly convective or stratiform precipitation alternate (e.g., Collier and Immerzeel, 2015).

The remaining section differs in structure from the temperature and humidity parts (Sect. 3.2 and 3.3) for two main reasons.
First, the time series of the monthly ACC gradients (not shown) does not exhibit an obvious pattern that could be interpreted in terms of climatic seasons like $T$ and water vapor (Figs. 5a and 8a), and also these measurements do not allow us to resolve the diurnal cycle. Secondly, the sonic ranger measurements have a higher uncertainty than $T$ and RH measurements. The focus here, thus, is less on temporal cycles but rather on assessing the measurements from different perspectives.

### 3.4.1 Mountain scale perspective

To scrutinize the absolute values of our measurements, the characteristic vertical precipitation distribution on tropical mountains outlined above can be elaborated. This distribution exists in all three glacierized massifs of East Africa (Rwenzori, Mount Kenya, Kilimanjaro), where the zone of maximum amounts typically lies between 2000 and 3000 m a.s.l. (Hastenrath, 1984). Figure 10 summarizes the results from multi-year measurement campaigns that targeted the vertical precipitation pattern on the southern slope of Kibo up to ~4500 m. Modeled precipitation on the southern slope is also
included in Fig. 10. Although the different sources are not comparable in a quantitative-statistical sense, since they originate from different locations and cover varying time periods, they demonstrate some consistent features of the mountain's hydroclimate. Figure 10 underlines there is a distinct belt of maximum precipitation around 2000–2600 m a.s.l., above which the amounts drop sharply. Further, short-term variability is obvious through the R03 profile that shows uniform negative deviations from the other sources; R03 data were collected during a period (10/1999–09/2001) dominated by La Niña
conditions and widespread precipitation deficits in East Africa (Anyamba et al., 2002). Spatial variability within an elevation zone can also be large, as the denser data sets (AVG, model) demonstrate. In the summit zone, sonic ranger-derived ACC and modeled precipitation confirm very dry conditions at the peak with less than 500 mm per year above the crater rim, and 500 to 700 mm per year below the rim down to 5000 m altitude. The reasonable agreement between these two different data sources (model, SR50a) lends confidence in the usability and post-processing of the sonic ranger measurements.




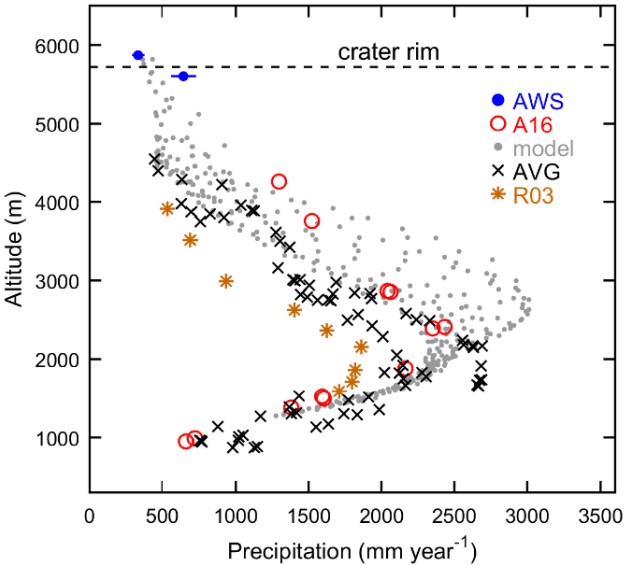

**Figure 10: Mean annual precipitation or (AWS) accumulation versus altitude at various locations in the southern mountain sector from different sources: AWS data and all model grid cells on the southern slope for 10/2009–09/2013; published values from Appelhans et al. (2016), A16 (2012–2014); Røhr and Killingtveit (2003), R03 (10/1999–09/2001); and again Appelhans et al. (2016) who constructed a longer-term average (AVG) since 1997 from the network of Hemp (2006). The error bar for AWS data (Table 1) is hardly visible at that scale. Horizontal dashed line shows the approximate altitude of the crater rim.**

### 3.4.2 The local observation perspective

Regarding the measurements themselves, a look at the smallest resolved time unit (daily) seems reasonable to check whether our processing algorithm produces unrealistic ACC amounts on some days that contaminate the mean value statistics. Figure

11 shows that this concern is not warranted. In both seasons the typical range of events comprises a few to max. 10 cm per day, which are magnitudes that we experienced during field work on snowfall days, and that also resulted from the manual deduction of ACC (not using the algorithm presented in this paper) from the sonic ranger record (Mölg et al., 2009a, 2009b). Thus, the daily ACC amounts in Fig. 11 per se do not seem unrealistic. What makes the big difference are the frequencies of certain conditions. The number of days in our four-year record, when more daily ACC falls at AWS4 than at AWS3 or when

AWS4 experiences ACC but AWS3 not, is clearly larger than for the converse situations.

Regarding other observational evidence that would support distinct ACC frequencies between AWS3 and AWS4, one relevant information was first expressed by Kaser and Osmaston (2002); it is the circumstance that daytime convective cloud growth from the base of the mountain upslope often encounters a sudden barrier right below the crater rim at ~5700 m. The

authors also hypothesized that the wide summit plateau of the mountain, covered by dark volcanic ashes (Fig. 1), plays a role by developing its own local thermal environment that prevents convergence of ascending air and clouds above the plateau, thus limiting deep convection. Such situations have formed one of Kilimanjaro's postcard-like idylls, where only the uppermost 100–200 m of the mountain appear from a "sea of cloud". Important from a scientific viewpoint, however, is that subsequent numerical atmospheric modeling found physical arguments that explain the cloud barrier, which is detailed

further below (Sect. 3.4.3). Thus the crater rim seems to be a kind of local moisture divide (and is therefore emphasized in





Fig. 10), and the Kaser and Osmaston (2002) description provides a basis for wet conditions at AWS4 and dry ones above at AWS3. For the opposite situation with more ACC at AWS3 than at AWS4, a result reported in Mölg et al. (2009b) holds relevance. They found a strong linear correlation between daily ACC at AWS3 and the free-tropospheric vapor content of the impinging large-scale flow. This relation indicates that precipitation amounts are not controlled by low-level water vapor

alone, giving the upper summit zone of Kibo a degree of influence detached from the processes along the slope (as discussed already in 3.3.2). From this perspective, occasional higher ACC at AWS3 (although clearly less frequent; see Fig. 11) seems plausible.

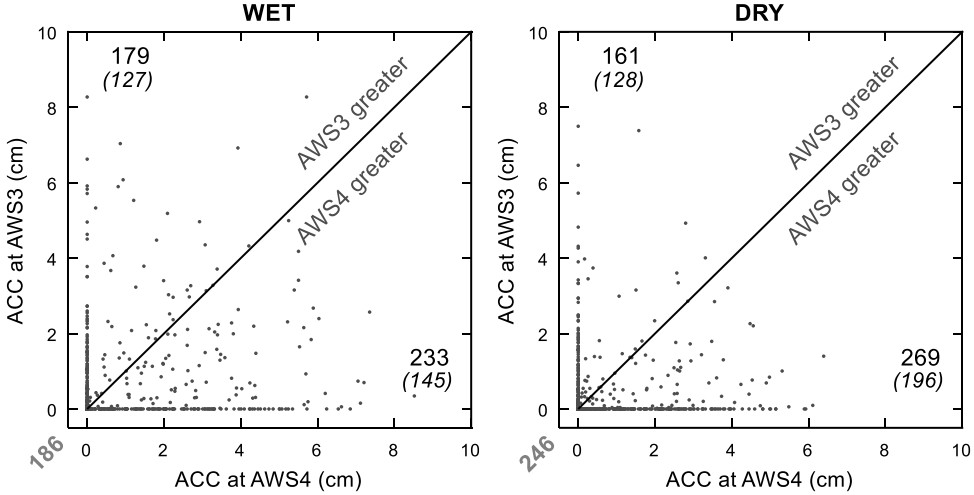

**Figure 11: AWS3 versus AWS4 daily accumulation (ACC) in all months of the (left) wet and (right) dry seasons between 8 October**
**2009 and 30 September 2013. Number of days when either AWS3 or AWS4 ACC was greater are displayed in the respective corners (number in parentheses are the contained cases when the other station recorded zero ACC). In grey below the lower left corner of the plots is the number of days when ACC is zero at both stations.**

A further possibility to assess the sonic ranger-derived ACC data is cross-checking with the independent humidity
measurements at the stations. Figure 12 stresses the early morning (left) and noon (middle) on all days when ACC at AWS4 was greater than at AWS3, i.e. on days that represent the most frequent precipitation condition for our problem. RH at the start of the day shows a strongly linear relation between the two stations, yet the course of the morning brings substantial change. By noon, much more data from AWS4 settle in a region of high RH (grey shading) than from AWS3. This disrupted linearity could only be the effect of the "usual" upslope transport of water vapor, as discussed Sect. 3.3. The more important
question is whether the days with greater ACC at AWS4 show a closer approach to saturation in the course of the day. The right panel of Fig. 12 would argue yes; it reveals that the RH excess at AWS4, relative to AWS3, is clearly higher on such days than on the days with greater ACC at AWS3. In particular, the differences between the two groups of days are statistically significant in the 9:00–14:00 LT window. This is exactly the time of day when regional atmospheric conditions are most influential on precipitation formation over Kibo (Mölg et al., 2009b). The RH measurements thus support the sonic-





ranger derived ACC events with their characteristic of being more frequent at AWS4, which is the main reason of the mean

ACC gradient. Figures 11/12 are based on Parameter Combination 1 and reference settings in the SR50a data treatment (see

Supplement, Table S1), but usage of the other settings leads to the same interpretations.

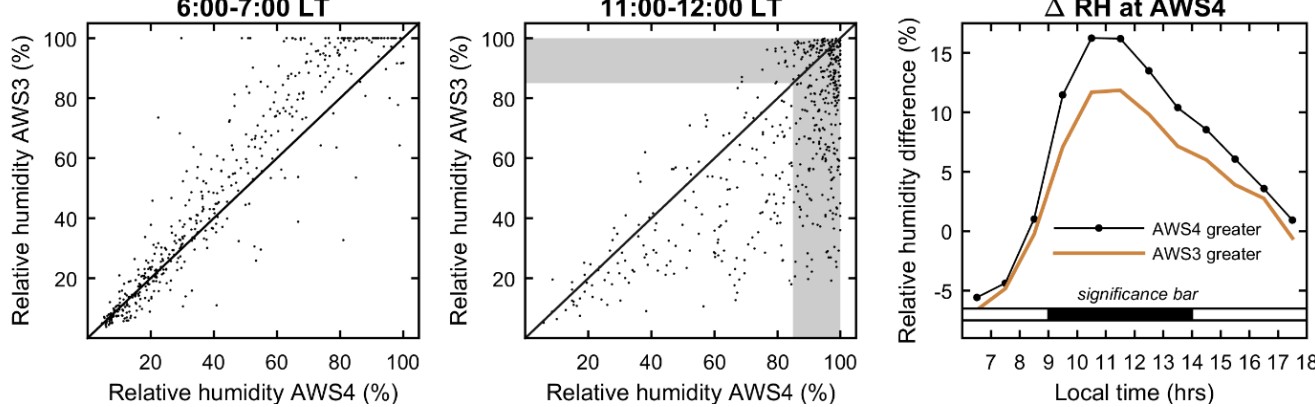

**Figure 12: AWS4 vs. AWS3 hourly mean relative humidity at (left) 6:00–7:00 LT and (middle) 11:00–12:00 LT for all days**
**between 8 October 2009 and 30 September 2013 when solid precipitation at AWS4 was greater than at AWS3 ($n$ = 502 days; cf.**
**233 + 269 in Fig. 11). Shading indicates RH > 85% to facilitate discussion in the text. Also, (right) the mean diurnal daytime cycles**
**of the RH excess at the lower station ($\Delta RH = RH_{AWS4} - RH_{AWS3}$) on the same days (black line) and on all days with the opposite**
**condition (brown; solid precipitation at AWS3 greater than at AWS4). Horizontal bar above the x axis identifies the hours when**
**differences in $\Delta RH$ are significant at the 5% level (black fill) in both a two-tailed Wilcoxon rank sum and Kolmogorov-Smirnov**
**test.**

It is understandable that other measurements of the ACC gradient on high mountains in convective climates for very similar

conditions (equatorial, free-standing, 5–6 km altitude) are not available, yet we found little in the literature that is

comparable. A first assessment could still conclude that the reported mean ACC gradient, which translates to roughly 10 mm

w.e. month$^{-1}$ decrease per 100 m elevation, is not unexpected for high-elevation regions. Figure 10 suggests so for the 4–5

km altitude band, and measurements above 3 km during convective seasons in other high mountains yielded similar

magnitudes (e.g., Baral et al., 2014). The sheer number of the ACC gradient, therefore, is realistic.

### 3.4.3 The model perspective

Since the modeled and measured precipitation in the summit zone provide a consistent idea of the gradient (Fig. 10), the

basic analyses below target other model variables to explore whether the sharp precipitation decrease with elevation in the

summit zone is supported from a physical standpoint. First it might be useful to recall the fundamental mechanisms of

orographic precipitation formation briefly, a recent review of which is Kirshbaum et al. (2018). In general, moist convection

can be initiated by two mountain influences: mechanical forcing where the flow responds to the mountain as an obstacle, by

climbing over or detouring around it; and thermal forcing where the flow responds to differential surface heating/cooling, by

developing a thermally direct anabatic or katabatic circulation. In reality both forcings typically act together to strengthen or





weaken regions of air ascent/descent, yet separating both precisely is a very complex problem (Kirshbaum and Wang, 2014). Mechanical forcing of mesoscale air flow on Kilimanjaro was studied in detail by Mölg et al. (2009b), and they found that the main mechanical flow regime is always the "flow around" type even when impinging air masses show very different

vertical stabilities. Therefore, and also because the thermal circulation clearly emerged in the analyses for $T$ and water vapor gradients, focusing on the possible effects of thermal forcing variability on the observed ACC gradients makes most sense.

In the following we address standard diagnostics for a thermally forced environment (e.g., Behrendt et al., 2011; Dong et al., 2016), including parcel-based metrics (e.g., CAPE). Starting with the latter, the maximum CAPE on the southern mountain

slope is highest in the model for days when ACC at the upper station (AWS3) is greater than below (Fig. 13, left); differences to days with higher ACC at AWS4 are especially obvious (and statistically significant) in the afternoon. Furthermore on such days (more ACC at AWS4 than at AWS3), the equivalent potential temperature ($\theta_e$) sees a significant lowering already at 9:00 LT in the entire upper half of the mountain slope (Fig. 13, middle). By afternoon (black curve), when $\theta_e$ typically increases over the entire mountain, the differences are mostly weakening except in the uppermost

elevations where they intensify (and remain statistically significant); the steepest gradient now exists between the AWS4 and AWS3 altitudes (Fig. 13, middle), which suggests a massive reduction in moist entropy toward the summit on days when moister conditions occur at AWS4. The differences in the water vapor flux on these days (Fig. 13, right) portray a more sophisticated picture. At 9:00 LT in the morning, there is a monotonically smaller flux above ~3000 m with high significance in the summit zone, suggesting a weaker upper tip of the thermal circulation in the morning. At 13:00 LT (black curve) the

pattern is more variable and there are no important differences in the uppermost elevation bands. However, a higher vapor flux centered in the 4600 ± 300 m altitude (Fig. 13, right) is an interesting feature that will be subject of discussion in the next paragraph. All the mentioned differences – CAPE, $\theta_e$, and water vapor flux – intensify when we consider the extreme subgroup of "AWS4 greater" days, which is precipitation at AWS4 only (and none at AWS3). A vivid example is included in the left panel of Fig. 13, where the CAPE reductions relative to days with more ACC at AWS3 are statistically significant

almost all day.



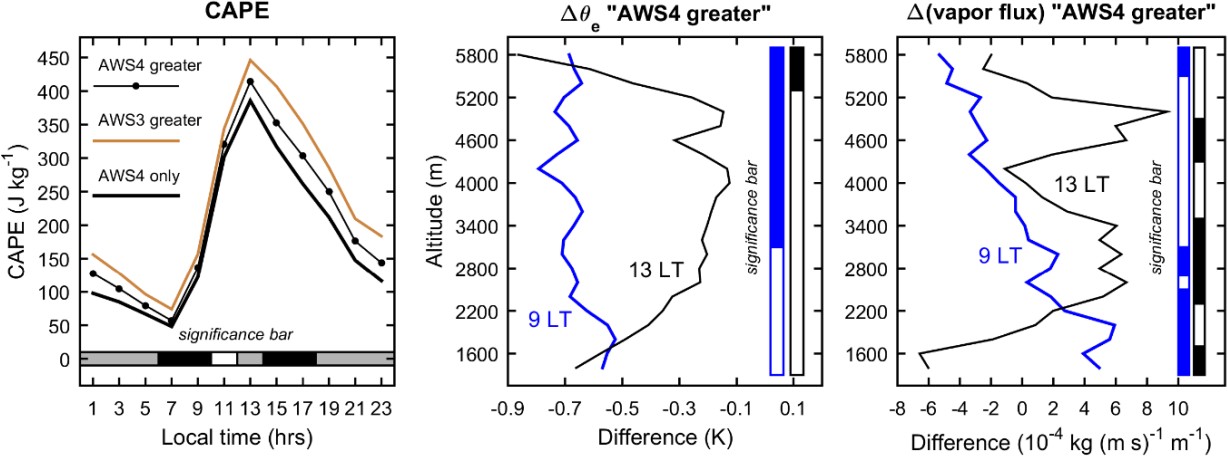

**Figure 13: Modeled differences between days with higher solid precipitation measured at AWS4 or AWS3 between 8 October 2009 and 30 September 2013 on Kibo's southern slope. (Left) Mean diurnal cycle of the local maximum CAPE for days with greater amounts at AWS4 than AWS3 ($n = 502$ days), the opposite case ($n= 340$ days), and precipitation at AWS4 but none at AWS3 ($n = 341$ days). (Middle and right) Mean differences in 2 m $\theta_e$ and (absolute) vertical water vapor flux, "AWS4 greater" days minus "AWS3 greater" days, for 9 LT (blue) and 13 LT (black) versus elevation (200 m bins). Horizontal and vertical bars in all panels identify the hours when differences are significant at the 5% level (fill) in both a two-tailed Wilcoxon rank sum and Kolmogorov-Smirnov test; on the left this bar distinguishes between "AWS 4 greater" vs. "AWS3 greater" (black fill) and "AWS3 greater" vs. "AWS4 only" (grey fill; partly covered by the black fill).**

The above features of the model's thermal circulation support the main difference unraveled from measurements, namely a group of less frequent days with higher ACC at AWS3, and a group of more frequent days with higher ACC at the lower-lying AWS4; the latter are favored over the former by reduced CAPE on the southern slope, reduced moist entropy at high altitude, and reduced water vapor fluxes in the summit zone. In-depth studies of atmospheric dynamics on low-latitude mountains also underlined the influence of the thermal circulation on the precipitation distribution. Wang and Kirshbaum (2015) and Junquas et al. (2017), for instance, demonstrated that precipitation over the mountains decreased substantially when latent heating in the clouds or surface heat fluxes were turned off in their models. At this point we should also recall published work that studied the physics of Kilimanjaro's mesoscale circulation in depth. This work delivered an explanation for the Kaser and Osmaston (2002) observation of clouds hanging at the crater rim frequently, and thus for the existence of a strong moisture gradient in the summit zone. Mölg et al. (2009b) showed that the halt of vertical cloud growth right below the summit plateau results from a frequently observed upstream profile with potentially stable flow at 700–800 hPa. Mölg and Kaser (2011) presented the result that such cases develop a statically stable region around 635 hPa in the cloud, constraining convective dynamics. Amid these former findings, the increased water vapor flux between 4300 and 4900 m in the afternoon on days when more ACC falls at AWS4 (Fig. 13, right) points to the formation of a stable layer.



## 4 Conclusions

Four years of overlapping AWS measurements at 5603 and 5873 m on Kersten Glacier (2009–2013) provide a great opportunity to extend our empirical knowledge of elevation gradients in temperature, humidity, and precipitation to a special
environment, characterized by its equatorial location and a massive free-standing mountain, Kilimanjaro. There are at least two remarkable points that fuel the knowledge extension in comparison to previous studies of gradients on (mostly, mid-latitude) glaciers.

First and at the local scale, the measured $T$ and ACC gradients seem comparatively strong, with average decreases of 0.75 K
and 114 ± 16 mm w.e. year$^{-1}$ per 100 m altitude, respectively. Mean RH, by contrast, differs hardly between the two measurement sites, a result that agrees with the observation of near-constant RH with elevation on extratropical mountain glaciers (e.g., Greuell et al., 1997). What is moreover striking in the formation of the mean $T$ gradient is a variability between wet and dry seasons that is anchored in a large diurnal cycle. $T$ gradients are roughly two times (dry season) or three times (wet season) as strong in the early morning as in the late afternoon. Similarly, the elevation decrease in water vapor
pressure fluctuates within a day and is ~4 times as large around noon as during the night. Measurement accuracy precludes resolving a diurnal cycle in ACC, but the measurements suggest that the negative elevation gradient is mainly due to a higher precipitation frequency at 5603 m than at 5873 m, rather than due to larger amounts at the lower site during coincident precipitation events.

The second point embodies the finding that the measured local gradients are largely constrained by the mesoscale atmospheric circulation over the mountain. This point emerged with the question after the reason for the distinct diurnal cycles in measured $T$ and vapor pressure gradients, in combination with the well-timed availability of data from a high-resolution mesoscale atmospheric model study (Collier et al., 2018) covering the four-year measurement period. The model output reliably shows the establishment of a thermally direct circulation over the mountain in response to diabatic surface
cooling and heating, with downslope flow during night and upslope flow during the day. Such flow patterns are well known from other tropical mountains, and mountain valley flows in general, based on both observations and numerical modeling (e.g., Wang and Kirshbaum, 2015; Giovannini et al., 2017). On Kilimanjaro, warm-air advection begins with the onset of solar insolation and is directed upslope until the late afternoon; since it intensifies with altitude, a continuous reduction of $T$ gradients in the summit zone is observed simultaneously. Moisture is also pumped upslope with the daytime circulation from
the rainforest belt, yet the time window of water vapor increase in the summit zone is limited to roughly 9:00–13:00 LT and the rates are stronger at 5600 m than above. Regarding the precipitation decrease with elevation, the model corroborates the observations by showing that reduced CAPE on the southern slope, together with a strong decline in moist entropy and reduced water vapor fluxes in the summit zone, hamper convective activity at the highest elevations on days when ACC fell at AWS4 but less or no ACC was recorded close to the summit at AWS3.


Local influences on the measured cycles of $T$ and vapor pressure gradients operate to a lesser extent, yet cannot be ruled out entirely. The energy balance in the summit zone shows a consistent contribution to the nighttime steepening of $T$ gradients and the intensification of this process in the wet season. Further, the local sublimation on Kersten Glacier most likely initiates the measured onset of the vapor pressure gradient weakening right after 12:00 LT; this weakening begins later in the model's mesoscale circulation that does not resolve differential sublimation on the glacier. At the other side of the scale spectrum, diagnosis of the surface wind vectors in the model suggests that the very highest elevations on Kilimanjaro (including our upper measurement site) feel relatively little impact from the mesoscale circulation but more from the large-scale geostrophic flow. This complements previous studies that found a clear relationship between the AWS3 data from 5873 m and the surrounding free troposphere at 500 hPa (Mölg et al., 2009a, 2009b). All together, these findings again underscore that Kilimanjaro is a fantastic site for the study of scale interactions in the climate system (e.g., Mölg et al., 2012).

Recalling the introduction of this article and the use of meteorological elevation gradients in glacier and other land surface models, the null hypothesis that the often-used $-0.65$ K $(100$ m$)^{-1}$ is an adequate description of the high-altitude glacier zone on Kilimanjaro must be rejected. Previous studies for other mountain regions also rejected this specific gradient (Minder et al., 2010). Usage of constant RH and a negative precipitation gradient for high-altitude tropical glaciers, however, is supported by the AWS data. Careful thought should nevertheless be invested whether a high temporal resolution in such models (e.g., hourly) would allow implementation of the obvious diurnal fluctuations in $T$ and water vapor gradients or different frequencies of ACC events at different elevations. With the case study of a high-altitude equatorial glacier on a free-standing mountain, our results widen the literature body on meteorological elevation gradients. Moreover, our study design provides a framework for future work to systematically examine mesoscale flow influences on local elevation gradients in glacierized mountains.

**Data Availability.** The AWS data from Kilimanjaro are available upon request from the authors, since they represent a very specific data set. The other employed data sets were already published previously.

**Author contributions.** TM designed the study and wrote the paper. TM, EK, and CS analyzed the data. TM, NC, DH, GK, RP, and MW conducted field work on Kilimanjaro, with MW, RP, and GK leading the AWS4 campaign. EC led the
atmospheric modeling. All authors continuously discussed the results, analysis design, and the paper preparation.

**Acknowledgements.** Measurements at AWS3 and AWS4 were supported by various funding agencies over the years: first of all the FWF (Austrian Science Fund), but also the Tyrolean Science Foundation and the Alexander von Humboldt





Foundation. Recent funding of Kilimanjaro research was from the DFG (German Science Foundation) with grants MO
2869/1-1 and MO 2869/3-1. We thank Tim Appelhans for providing rainfall data and for discussion.

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
