# Peer review of "Mesoscale atmospheric circulation controls of local meteorological elevation gradients on Kersten Glacier near Kilimanjaro summit"

_Earth System Dynamics, 2019_

## Referee Comment (RC1) · Anonymous Referee #1 · 19 Jan 2020

Comments on the "Mesoscale atmospheric circulation controls of local meteorological elevation gradients on Kersten Glacier near Kilimanjaro summit" by Mölg et al.

Using a subset of AWS data observed at 2 stations on Kersten Glacier, Kilimanjaro, and high spatial resolution simulation results, the authors produce estimates of vertical gradients of air temperature, humidity, and accumulation. With the added value of high spatial resolution simulation, the authors analyze the mesoscale circulation controls of meteorological vertical gradients. Based on that analysis, the authors make recommendations as to what gradient values are most appropriate for the glacier zone on Kilimanjaro. The manuscript has some strong points: the subject is relevant, some of

the methods are well established and, perhaps most importantly, the manuscript deals with a region where observations are relatively rare. Overall, the manuscript reads well and almost ready for publication, with a few technical aspects to correct and a few minor points to consider.

Minor comments:

Please provide more information about the configuration of atmospheric model experiments, for example, the name of the model, microphysics, convection schemes, land surface or glacier surface schemes. Just providing a reference is not convenient enough for readers.

Line 9: "who detail climatological signatures" → "who detailed climatological signatures".

Figure 3: How large is the "southern mountain slope" region or how many grid points?

Figure 5b: Please plot the simulated mean diurnal cycle of air temperature gradient in Fig 5b as well. The computation of the standard deviation may be overplotted in Fig 5b.

Line 368: "Figs. 5b and S2" → "Figs. 5b and S3".

Line 536: "the mountain's hydroclimate" → "the hydroclimate of the mountain".

Line 566: "the Kaser and Osmaston (2002) description" → "the Kaser and Osmaston (2002)'s description".

———————————————————

---

## Referee Comment (RC2) · Anonymous Referee #2 · 30 Apr 2020

**Review of ESD-Manuscript as a Referee: Joseph Romanus Mukabana**

**Date:13[th] April 2020**

**Journal:** ESD
**Title:** Mesoscale atmospheric circulation controls of local meteorological elevation gradients on Kersten Glacier near Kilimanjaro summit
**Author(s):** Thomas Mölg et al.
**MS No.:** esd-2019-51
**MS Type:** Research article

**GENERAL COMMENTS**

This section addresses the overall quality of the paper based on my evaluation and assessment as a referee.

Sequence to the given review criteria and general obligation for referees, and further having read the manuscript, I would like to state that in my considered opinion:

The authors of this paper have addressed relevant scientific questions related to their topic and scope of their chosen study. The paper outlines novel concepts and ideas including a field campaign / experiment in a complex high altitude environment. In my view, the results are sufficient to support the interpretations and conclusions reached in this study.

As a case study of a high-altitude glacier on a mountain located in the tropics and specifically in the equatorial zone, the authors arrived at substantive conclusions showing that the results augment similar studies on meteorological elevation gradients and also provide a framework that could be used to examine the influence of mesoscale flow on mountains with glaciers. It this study, the authors displayed an amazing teamwork and division of labour that included designing the study, writing the paper, analyzing the data, conducting field work and undertaking the field campaign.

I find the scientific methods used and the assumptions made in this study valid and clearly outlined. The authors description of the field experiments and calculations are fascinating and I believe they are sufficient and precise to allow their reproduction by fellow scientists. The authors took an incisive literature review relevant to the study shown in cited references and also have given proper credit to related work in the body of the text. Moreover, the title of the study reflects the content of the paper and the abstract provides a concise summary of the work accomplished.

The overall presentation of the paper is well structured and the language used is fluent and precise. The mathematical formulae, symbols, abbreviations, and units in the manuscript are correctly defined and properly used and most parts of the paper have clarity and need not be combined or eliminated. The number and quality of references as well as the amount and quality of supplementary material are appropriate for this manuscript.

**SPECIFIC COMMENTS**

This section addresses individual scientific issues in the manuscript.

1. **Introduction**

It is stated here that this four-year study (2009-2013) used elevation gradients in basic meteorological variables as model parameters on a chosen domain (5603m and 5872m) on Kersten Glacier of Mt Kilimanjaro– to construct spatially distributed atmospheric drivers of glacier mass balance. It is also stated that if the mass balance model was coupled to an atmospheric model, the coupled model would deliver these parameters at different elevations (e.g. Collier et al 2013).

**Question:** spatial model resolution allowing (say resolution of 400m or less) and if computational costs were availed, do you think that it would have been better to proceed with a coupled model study, whereby you would see the interaction of the mesoscale systems (local circulation) with regional scale (synoptic scale) systems right away? Or was the lack of information on observed elevation gradients for tropical glaciers a motivation for you to undertake this study that involved field experiments?

2. **Methods and Data**

It is sated that automatic weather stations (AWS3 at 5873m and AWS4 at 5603m) were placed on Kersten Glacier with intent to gain knowledge of elevation gradients in the basic meteorological variables.

**Question:** Do you think installation of additional AWS (say, AWS5) at a lower altitude (still close to AWS4, but below and out of the Kersten Glacier zone) would have acted as some kind of *Control* to measure gradients out of the glacier but up-close; just for comparison?

2.3 **Processing of the sonic ranger records**

It is indicated that, "*The SR0a records from Kibo must be carefully post-processed due to intervals of noise data.*" Also, "*Due to the noise, we usually did not attempt to interpret the hourly data but daily values at the most.*"

**Question:** Why were the sonic data so important that you had to find a way of applying them (from hourly to daily values) despite the noise? The significance of sonic data is not well explained – what is the advantage of using sonic data instead of other measurement methods?

2.4 **Atmospheric model experiment**

It is sated that Collier et al (2018) with a spatial resolution of 800m showed that the model reproduces the important features of atmospheric conditions in the summit zone of Kibo (Fig 3). But Kersten Glacier is only represented by one grid cell in this model despite its high resolution. This means, therefore that evaluating simulated gradients on the glacier scale would not be possible.

**Question:** In your opinion, would it have been appropriate for Collier et al (2018), in designing the model simulations, to take into account the fact that they were dealing with a study of batch of receding glacier (and not expansive ice sheets as found in mid-latitudes) and needed to further increase the model resolution over Kilimanjaro to say 200m – 250m ?

**3.0 Results and discussion**

It is mentioned here that all elevation gradients are per 100m, which seems practical with regard to the altitude extent of most present mountain glaciers (expansive ice sheets?) and distinguishes from micrometeorological  gradients that are usually given per meter.

**Question:** Considering that the Kersten Glacier is just one significant block of remaining ice on Mt Kilimanjaro, don't you think that the limited area it occupies, and which formed your study domain, could be attributable to the creation of micrometeorological gradients and would thus qualify to have elevation gradients measured per meter?

**3.2.2 Non local control factors**

It is stated here that, "*These features together indeed suggest that mesoscale circulation communicates the heating signal from the lowlands to higher elevations over the course of the day and mixes potentially warm air upslope; the intensification of this process with altitude helps to explain why daytime elevation T gradients in the summit zone are reduced (Figs 5b & S2).*" Italics mine.

**Question:** The glaciers on MT Kilimanjaro, Mt Kenya as well as Mt Ruwenzori have decreased substantially over time. This glacier recession has been attributed to global warming and attendant impacts of climate change. In your considered opinion, can the vertical heat fluxes from lowlands and their advection to higher elevation, as explained in your statement, be linked to the recession or decline of glaciers on tropical mountains in the equatorial zone?

**III. TECHNICAL CORRECTIONS**

This section addresses typographical errors, like spelling or sentence reconstruction, etc.

**Section 3. Results and discussion**

a)  Paragraph in 250 starting with, "*In the following we report all elevations ………….*"
    Since 3 is a section, I reckon that what follows is/are sub-sections, therefore would it not be appropriate if we put the word "*subsection (s)*", so that the sentence would read, *"In the following subsections, we report……."*

b)  In the same paragraph under Section 3, there is need to recast the last sentence, which reads "*The derived quantity should thus be a good proxy of solid precipitation, yet we will continue to call it accumulation (ACC) to not forget that the measurement principle differs from a regular precipitation gauge.*" Italics mine.
    → The sentence could read, "*Although the derived quantity is a good proxy of precipitation, we will continue to refer to it as accumulation (ACC) and note that the measurement principle of ACC differs from that which applies regular precipitation gauges.*"
    →
c)  Under 385, note the sentence, *"…….. a tendency that is maintained for rest of the night and peaks at 7:00LT."* Please put the word "*the*" after "for" to read, *"…….. a tendency that is maintained for the rest of the night and peaks at 7:00LT."*

END

---

## Author Response (AR1)

*Dear Editor,*

*Thank you for inviting us to submit the revised version of our manuscript. For the point-by-point reply below we take the text from our final response as the basis (green) and indicate in yellow where changes in the revised MS were announced, and in orange-shaded boxes below how we implemented them or what other changes we made. We hope this will allow you to comprehend the revision actions easily. --- Please note that no revised Supplement is attached as no changes were requested.*

*Thanks for your consideration.*

*Sincerely,*
*Thomas Mölg & co-authors*

**Referee #1 (anonymous)**

*RESP: Thank you very much for the compliments in your general comment.*

**Minor comments:**
Please provide more information about the configuration of atmospheric model experiments, for example, the name of the model, microphysics, convection schemes, land surface or glacier surface schemes. Just providing a reference is not convenient enough for readers.
*RESP: In the revised MS, we will direct the reader to the relevant details explicitly. We would prefer, however, to not include such a table, since these tables appear in all publications that presented the modeling under question (Mölg & Kaser, 2011; Mölg et al., 2012; Collier et al., 2018) and the present paper is not contributing any new model data. We suggest to change the text in Section 2.4 and make it clear for the reader where the model setup can be looked up.*

**CHANGES MADE**: In the first paragraph of Section 2.4., we added a sentence detailing where the other important model settings can be found, and what options they include.

Figure 5b: Please plot the simulated mean diurnal cycle of air temperature gradient in Fig 5b as well. The computation of the standard deviation may be overplotted in Fig 5b.
*RESP: We are sorry to say that it is not clear to us why this referee would like to see the simulated diurnal cycle and the standard deviation (of hourly observations? Between model and observations?) in Fig. 5b (air temperature), but not so in Figure 8b which is the analog case for vapor pressure. Our attempt with these two figures was to show the observations in the paper, and the same for the model in the supplement to maintain a clear order. --- Regarding standard deviation, we did these plots during the research (see below for air temperature) but decided to remove them for the final manuscript since the result did not reveal any substantial new insight.*

[Figure]

*Figure R1: as Figure 5 in the paper, but with standard deviation of hourly gradients in panel b).*

Figure 3: How large is the "southern mountain slope" region or how many grid points?
RESP: We will provide the information in the revised MS in Section 2.4 where we define the area (the answer is 350 grid points).

**CHANGES MADE:** We now give the number (350) in the caption for Figure 3b.

Line 119: "who detail climatological signatures" ! "who detailed climatological signatures".
Line 368: "Figs. 5b and S2" ! "Figs. 5b and S3".
Line 536: "the mountain's hydroclimate" ! "the hydroclimate of the mountain".
Line 566: "the Kaser and Osmaston (2002) description" ! "the Kaser and Osmaston (2002)'s description".
*RESP: We will check and correct these cases.*

**CHANGES MADE:** We corrected 119, 368, and 536. We left 566 unchanged since the three native English speakers among the authors prefer the original writing.

**Referee #2 (Joseph Romanus Mukabana)**

**General comments:**

…
*RESP: Thank you very much for the compliments.*

**Specific comments:**

**1. Introduction**

It is stated here that this four-year study (2009-2013) used elevation gradients in basic meteorological variables as model parameters on a chosen domain (5603m and 5872m) on Kersten Glacier of Mt Kilimanjaro– to construct spatially distributed atmospheric drivers of glacier mass balance. It is also stated that if the mass balance model was coupled to an atmospheric model, the coupled model would deliver these parameters at different elevations (e.g. Collier et al 2013).

**Question:** spatial model resolution allowing (say resolution of 400m or less) and if computational costs were availed, do you think that it would have been better to proceed with a coupled model study, whereby you would see the interaction of the mesoscale systems (local circulation) with regional scale (synoptic scale) systems right away? Or was the lack of information on observed elevation gradients for tropical glaciers a motivation for you to undertake this study that involved field experiments?

*RESP: Dr. Mukabana is correct, the motivation was indeed the lack of information on observed elevation gradients. It was simply a lucky circumstance that the model data were available in addition. But since the modeling took place two years earlier, these experiments were not designed specifically with regard to the measured gradients. In our previous Kilimanjaro papers, we published selected details assessing the interaction of mesoscale and large-scale factors (e.g. Collier et al., 2018). As the glaciers are very small these days, we do not have any evidence that they would feed back to the atmosphere significantly, thus interactive coupling is of less importance in the modeling framework for Kilimanjaro. For your interest (but out of paper scope), we made a relevant test in the study Mölg et al. (2012), where we removed the glaciers on the summit in the model and did not obtain strong effects on the atmosphere over the mountain slope.*

**2. Methods and Data**

It is sated that automatic weather stations (AWS3 at 5873m and AWS4 at 5603m) were placed on Kersten Glacier with intent to gain knowledge of elevation gradients in the basic meteorological variables.

**Question:** Do you think installation of additional AWS (say, AWS5) at a lower altitude (still close to AWS4, but below and out of the Kersten Glacier zone) would have acted as some kind of *Control* to measure gradients out of the glacier but up-close; just for comparison?

*RESP: This is a valuable thought and there is no doubt that having such a hypothetical AWS5 would have helped for interpreting the data. The truth is that access to the lower elevations of Kersten Glacier (or even below its terminus) is very difficult and, therefore, AWS4 was the best we could achieve.*

**REMARK:** Please note that we emphasize the difficult access in the paragraph staring in line 128, so it should be clear to readers that installation of multiple AWSs was not possible.

**2.3 Processing of the sonic ranger records**

It is indicated that, "*The SR0a records from Kibo must be carefully post-processed due to intervals of noise data.*" Also, "*Due to the noise, we usually did not attempt to interpret the hourly data but daily values at the most.*"

**Question:** Why were the sonic data so important that you had to find a way of applying them (from hourly to daily values) despite the noise? The significance of sonic data is not well explained – what is the advantage of using sonic data instead of other measurement methods?

*RESP: This is a fair argument, and we will state the reason explicitly in the revised MS. It is a technical issue to operate standard rain gauges at this elevation (mostly concerning power supply), thus SRS is the alternative. Since it is impossible to fully correct for the effect of rapidly changing air temperature profiles on sonic velocity, hourly data come with a too high uncertainty.*

**CHANGES MADE**: We modified and added text in the first paragraph of Section 2.3 to explain why sonic ranger sensors are widely used in cold regions. --- The issue of noise and the role of the sonic velocity (as an argument why we use daily instead of hourly data) were already explained in detail in the second paragraph of the same section, which we maintain in the revised MS.

**2.4 Atmospheric model experiment**

It is sated that Collier et al (2018) with a spatial resolution of 800m showed that the model reproduces the important features of atmospheric conditions in the summit zone of Kibo (Fig 3). But Kersten Glacier is only represented by one grid cell in this model despite its high resolution. This means, therefore that evaluating simulated gradients on the glacier scale would not be possible.

**Question:** In your opinion, would it have been appropriate for Collier et al (2018), in designing the model simulations, to take into account the fact that they were dealing with a study of batch of receding glacier (and not expansive ice sheets as found in mid-latitudes) and needed to further increase the model resolution over Kilimanjaro to say 200m – 250m ?

*RESP: Please see our response on Page 2 to your introduction comment. (i) Collier et al.'s (2018) experiments were designed with regard to examining ENSO and Indian Ocean influences on the mountain, for which 800 m was an adequate choice. (ii) The very small size of the glaciers excludes that resolving an atmospheric model grid further would alter any findings for the question in item (i). We corroborated the small influence of the glaciers on the meteorological conditions in the summit zone in the study of Mölg et al. (2012).*

**CHANGES MADE:** We decided to add a sentence in lines 247-249 to explain why 800 m was the choice in the previous model studies. --- In the second last paragraph of the conclusions section, we now also indicate the small effect of glacier existence on the climate of the high-elevation slopes.

**3.0 Results and discussion**

It is mentioned here that all elevation gradients are per 100m, which seems practical with regard to the altitude extent of most present mountain glaciers (expansive ice sheets?) and distinguishes from micrometeorological gradients that are usually given per meter.

**Question:** Considering that the Kersten Glacier is just one significant block of remaining ice on Mt Kilimanjaro, don't you think that the limited area it occupies, and which formed your study domain, could be attributable to the creation of micrometeorological gradients and would thus qualify to have elevation gradients measured per meter?

*RESP: It is an interesting perspective, but we do not think so. Our choice to use "per 100 m" was mostly driven by the existing literature on gradients over mountain glaciers. We argue in the paper (four items from lines 331-335) why micrometeorological influences are most probably small for elevation (vertical) gradients. Yet the referee's comment implies that there will be horizontal micrometeorological gradients. This is correct (yet not the topic of the present paper) and was picked up in several of our previous studies (e.g., Winkler et al. (2009), Erdkunde 64, 179-193).*

**3.2.2 Non local control factors**

It is stated here that, "*These features together indeed suggest that mesoscale circulation communicates the heating signal from the lowlands to higher elevations over the course of the day and mixes potentially warm air upslope; the intensification of this process with altitude helps to explain why daytime elevation T gradients in the summit zone are reduced (Figs 5b & S2).*" Italics mine.

**Question:** The glaciers on MT Kilimanjaro, Mt Kenya as well as Mt Ruwenzori have decreased substantially over time. This glacier recession has been attributed to global warming and attendant impacts of climate change. In your considered opinion, can the vertical heat fluxes from lowlands and their advection to higher elevation, as explained in your statement, be linked to the recession or decline of glaciers on tropical mountains in the equatorial zone?

*RESP: The overwhelming evidence from our research points to the fact that the reduction in local snowfall at the high elevations of Kilimanjaro is the major driver of modern glacier recession (and is related to global warming through influences on Indian Ocean dynamics). The sensitivity to local air temperature is limited, which is also true for Mount Kenya (e.g., Prinz et al., 2016). It is therefore unlikely that the heat pump mechanism described in the paper can be linked directly to glacier mass loss. --- If you are interested in the attribution of glacier recession, we invite you to look at the summary document available at <http://thomasmoelg.info/factsheet_kili.pdf>. --- In the revised MS, we will add a short text in Section 2.1 (where we introduce the research context) about these general findings.*

**CHANGES MADE:** We added a brief summary of 1.5 sentences in the first paragraph of Section 2.1, to explain the main findings on glacier loss and climate on Kilimanjaro for the interested reader. Due to this addition, we deleted the footnote on page 3, which was directed to the same information.

**III. TECHNICAL CORRECTIONS**

This section addresses typographical errors, like spelling or sentence reconstruction, etc.

**Section 3. Results and discussion**

a) Paragraph in 250 starting with, "*In the following we report all elevations* ............"
Since 3 is a section, I reckon that what follows is/are sub-sections, therefore would it not be appropriate if we put the word "*subsection (s)*", so that the sentence would read, "*In the following subsections, we report........*"

b) In the same paragraph under Section 3, there is need to recast the last sentence, which reads "*The derived quantity should thus be a good proxy of solid precipitation, yet we will continue to call it accumulation (ACC) to not forget that the measurement principle differs from a regular precipitation gauge.*" Italics mine.
   → The sentence could read, "*Although the derived quantity is a good proxy of precipitation, we will continue to refer to it as accumulation (ACC) and note that the measurement principle of ACC differs from that which applies regular precipitation gauges.*"
   →

c) Under 385, note the sentence, "*........ a tendency that is maintained for rest of the night and peaks at 7:00LT.*" Please put the word "*the*" after "for" to read, "*........ a tendency that is maintained for the rest of the night and peaks at 7:00LT.*"

RESP: We will check and correct these cases.

**CHANGES MADE**: We corrected all three items a) to c).

- - - - - -

**GRAMMATICAL CORRECTIONS**: Upon re-reading we made the following minor corrections from a grammatical perspective (highlighted green in the manuscript) to eliminate ambiguity:

19: added the one word "recorded"
124-125: reworded half a sentence
566: "information" replaced by "idea"
571: "appear from" replaced by "penetrate"
574: two words changed to fix the comparison statement grammatically
581: replaced "in all months" by "during"